



# Oxygenated VOCs as significant but varied contributors to VOC emissions from vehicles

Sihang Wang[1,2], Bin Yuan[1,2,*], Caihong Wu[1,2], Chaomin Wang[1,2], Tiange Li[1,2], Xianjun He[1,2], Yibo Huangfu[1,2], Jipeng Qi[1,2], Xiaobing Li[1,2], Junyu Zheng[1,2], Qing'e Sha[1,2], Manni Zhu[1,2], Shengrong Lou[3], Hongli Wang[3], Thomas Karl[4], Martin Graus[4], Zibing Yuan[5*], Min Shao[1,2]

[1] Institute for Environmental and Climate Research, Jinan University, Guangzhou 511443, China

[2] Guangdong-Hongkong-Macau Joint Laboratory of Collaborative Innovation for Environmental Quality, Guangzhou 511443, China

[3] State Environmental Protection Key Laboratory of Formation and Prevention of Urban Air Pollution Complex, Shanghai Academy of Environmental Sciences, Shanghai 200233, China

[4] Department of Atmospheric and Cryospheric Sciences, University of Innsbruck, Innsbruck, Austria

[5] College of Environment and Energy, South China University of Technology, University Town, Guangzhou 510006, China

*Correspondence to: Bin Yuan (byuan@jnu.edu.cn) and Zibing Yuan (zibing@scut.edu.cn)


## Abstract:

Vehicular emission is an important source for volatile organic compounds (VOCs) in urban and downwind regions. In this study, we conducted a chassis dynamometer study to investigate VOC emissions from vehicles using gasoline, diesel, and liquefied petroleum gas (LPG) as fuel. Time-resolved VOC emissions from vehicles are chemically characterized by a proton-transfer-reaction time-of-flight mass spectrometry (PTR-ToF-MS) with high frequency. Our results show that emission factors of VOCs generally decrease with the improvement of emission standard for gasoline vehicles, whereas variations of emission factors for diesel vehicles with emission standards are more diverse. Mass spectra analysis of PTR-ToF-MS suggest that cold start significantly influence VOCs emission of gasoline vehicles, while the influences are less important for diesel vehicles. Large differences of VOC emissions between gasoline and diesel vehicles are observed with emission factors of most VOC species from diesel vehicles were higher than gasoline vehicles, especially for most oxygenated volatile organic compounds (OVOCs) and heavier aromatics. These results indicate quantification of heavier species by PTR-ToF-MS may be important in characterization of vehicular exhausts. Our results suggest that VOC pairs (e.g. $C_{14}$ aromatics/toluene ratio) could potentially provide good indicators for distinguishing emissions from gasoline and diesel vehicles. The fractions of OVOCs in total VOC emissions are determined by combining measurements of hydrocarbons from canisters and online observations of PTR-ToF-MS. We show that OVOCs contribute $7.7\% \pm 6.2\%$ of gasoline vehicles of the total VOC emissions, while the fractions are significantly higher for diesel vehicles (40-77%), highlighting the importance to detect these OVOC species in diesel emissions. Our study demonstrated that the large number of OVOC species measured by PTR-ToF-MS are important in characterization of VOC emissions from vehicles.



# 1. Introduction

Volatile organic compounds (VOCs) are important trace components in the troposphere, as important precursors of ground-level ozone (Shao et al., 2009) and secondary organic aerosol (SOA) (Seinfeld and Pandis, 2006;Kansal, 2009;Ziemann and Atkinson, 2012). As the result, it is particularly important to identify emission sources of VOCs in the atmosphere. Vehicular emission is an important source of VOCs in cities around the world (Liu et al., 2008;Parrish et al., 2009), contributing approximately 25% to total VOC emissions in China (Ou et al., 2015;Wu et al., 2016;Sun et al., 2018). In order to control atmospheric pollution in urban and surrounding regions, it is necessary to understand source profiles and emission characteristics of VOCs from vehicles.

Emissions of VOCs from vehicles have been investigated extensively from tunnel studies (Cui et al., 2018;Zhang et al., 2018;Song et al., 2020), on-road mobile measurements (Li et al., 2017), and chassis dynamometer tests (Guo et al., 2011;Wang et al., 2013;Yang et al., 2018). Previous studies demonstrated that fuel types of vehicles strongly impact VOC emissions. Aromatics along with other hydrocarbons are known as compounds with high emissions in exhausts of gasoline vehicles (Wang et al., 2013;Ly et al., 2020). Some carbonyl compounds contribute significantly to emissions of diesel vehicles, at fractions much higher than gasoline vehicles (Tsai et al., 2012;Qiao et al., 2012;Yao et al., 2015;Mo et al., 2016). Moreover, there are still a large number of unidentifiable compounds in diesel vehicles (May et al., 2014). Furthermore, VOC emissions significantly decreased for stricter emission standards (Cao et al., 2016). In order to reduce emissions of most primary pollutants, more stringent emission standards and after-treatment devices have been implemented. The emission standard of China VI has already been implemented in July of 2019 in a few key cities in China and in July of 2021 nationwide. The emission limits for various air pollutants emitted by vehicles are significantly lower under the China VI emission standard (Wu et al., 2017). With the continuous development of engine and exhaust after-treatment technologies, emission characteristics of VOCs from vehicles may change and need to be frequently


updated.

Oxygenated volatile organic compounds (OVOCs) were found to be an important

group in vehicle exhausts, accounting for more than 50% of the total VOC emissions
for diesel vehicles (Schauer et al., 1999;Yao et al., 2015;Mo et al., 2016). Traditionally,
VOCs are collected in the canister or Tedlar bags, and then analyzed by gas
chromatography-mass spectrometer/flame ionization detector (GC-MS/FID), mainly
reporting emissions of hydrocarbons (Wang et al., 2017;Qi et al., 2019). Previous work
usually collected 2,4-dinitrophenyhydrazine (DNPH) cartridges and analyzed using
high-performance liquid chromatography (HPLC) for carbonyls (aldehydes and
ketones), which are both time-consuming and prone to contaminations (Mo et al.,
2016;Han et al., 2019).

The large variability of VOC emissions under different engine activities or

driving conditions require characterization of vehicular emissions at higher time
resolution. Proton-transfer-reaction mass spectrometry (PTR-MS) has been used in a
number of studies for measurements of vehicle emissions. VOCs from vehicle exhausts
under various driving and operational modes were measured by PTR-MS onboard a
mobile laboratory (Zavala et al., 2006;Zavala et al., 2009). Drozd et al. (2016) used a
PTR-MS to emphasize the importance of cold start for vehicles, concluding that VOC
emissions during cold start were equal to a 200 miles distance of driving during hot
stabilized condition. Proton-transfer-reaction time-of-flight mass spectrometry (PTR-
ToF-MS) can provide more powerful detection of various VOCs, thanks to the
measurements of whole mass spectra and high mass resolution (Cappellin et al.,
2012;Yuan et al., 2017). More OVOC species could be quantified from the measured
mass spectra based on parameterization methods for sensitivity of instrument
(Sekimoto et al., 2017;Wu et al., 2020).

In this study, we applied a PTR-ToF-MS along with a suite of other instruments

to measure VOCs emitted from gasoline, diesel, and liquefied petroleum gas (LPG)
vehicles. We investigated emission factors from different fuel types and emission
standards for representative VOC species exhausted from these vehicles. We used the
dataset to analyze contributions of various VOC groups to total VOC emissions in



different types of vehicles.

## 2. Materials and methods

## 2.1 Tested vehicles and the chassis dynamometer study methods

In this study, we conducted chassis dynamometer measurements to investigate
VOC emissions from vehicles using gasoline, diesel, LPG as fuel. All gasoline vehicles
are light-duty-gasoline-vehicle (LDGV) with the emission standards from China I to
China VI, whereas diesel vehicles can be classified into light-duty-diesel-truck (LDDT),
middle-duty-diesel-truck (MDDT), heavy-duty-diesel-truck (HDDT), and bus
associated with emission standards of China III to China V. In addition, the test vehicles
using LPG are all taxis, which are under mandatory scrappage after 8 years of driving
in China; as a result only China IV and China V for LPG vehicles were tested. Among
the 38 vehicles we tested, a fraction of vehicles was measured several times, with a total
of 62 experiments measured. The detailed information for test vehicles is summarized
in Table S1 and Table S2.
The short transient driving cycle (GB 18285-2018, Figure S1a), as one of the
widely used test methods for vehicle emissions in China (Li et al., 2012;Wang et al.,
2013), was used for measurements of gasoline vehicles and LDDT, each running for
three to five times. The short transient driving cycle methods were initially adapted
based on emission regulations of the Economic Commission for Europe (ECE) cycle
(Yao et al., 2003), which is developed and used in European countries (Laurikko, 1995).
The short transient driving cycle consist of four conditions, namely idling, acceleration,
deceleration and uniform speed, as shown in Fig. S1. For the MDDT and HDDT, we
customized a step-by-step test method, in which the vehicle accelerates to 20 km·h$^{-1}$,
40 km·h$^{-1}$ and 60 km·h$^{-1}$ in sequence after the engine activates, keeping at 20 km·h$^{-1}$
and 40 km·h$^{-1}$ for 2 minutes, and 60 km·h$^{-1}$ for 1 minute, respectively (Fig. S1) (Li et
al., 2021;Liu et al., 2021;Liao et al., 2021). In addition, the cold start was tested for a
number of vehicles after a cold soak for more than 12 hours at ambient temperature
(20-25 °C) before engine started. The measurements of cold start are compared to
measurements of hot start after a ~10 minutes break for the vehicles after previous



measurement. More details about cold start and hot start in this campaign can be found
in Li et al. (2021).

A custom-built sampling and dilution system for vehicles combining online and

offline sampling techniques was used in this study. As shown in Fig. S2, a portable
emission measurement system (PEMS, SEMTECH-DS, Sensors. USA) was employed
to measure emissions of CO, $CO_2$, $NO_X$, and total hydrocarbon (THC) directly from the
tailpipe of vehicles. A custom-built dilution system (Li et al., 2021;Liao et al., 2021)
was used for dilution of vehicular emissions, achieving dilution ratios of 10-100 for
different vehicles. After dilution, $CO_2$ and CO were measured using a Li-840A
$CO_2$/$H_2O$ Gas Analyzer (Licor, Inc. USA) and a Thermo 48i-TLE analyzer (Thermo
Fisher Scientific Inc. USA), respectively. Measurements of $CO_2$ before and after the
dilution system was used to determine the dilution ratio for each test (see details in Fig.
S3).

## 2.2  VOC measurements using PTR-ToF-MS

In this study, a Proton Transfer Reaction Quadrupole interface Time-of-Flight

Mass Spectrometer (PTR-QiToF-MS) (Ionicon Analytik, Innsbruck, Austria) with
$H_3O^+$ chemistry was used to measure VOCs (Sulzer et al., 2014). The mass spectra of
PTR-ToF-MS was recorded every 1 s as to capture characteristics of VOC species from
vehicle exhausts in real-time. Background measurements of the instrument were
performed using sampled air through a custom-built platinum catalytical converter
heated to 365 °C for 30 s before vehicle starts in each test. The more detailed setting
parameters for the instrument can be found elsewhere (Wu et al., 2020;Wang et al.,
2020a;He et al., 2022). Data analysis of PTR-ToF-MS was performed using the Tofware
software package (version 3.0.3, Tofwerk AG, Switzerland) (Stark et al., 2015).

A 23-component gas standard (Linde Spectra) was used for daily calibration of

PTR-ToF-MS during the campaign. VOC sensitivities from automatical calibrations
indicated quite stable instrumental performance for most of the VOC species (Fig. S4).
Another gas standard with 35-component VOCs (Apel Riemer Environmental Inc.) was
used for calibrations during the later period of this campaign to include more VOC





species in the calibration. The Liquid Calibration Unit (LCU, Ionicon Analytik,
Innsbruck, Austria) was used to calibrate a total of 11 organic acids and nitrogen-
containing species (Table S3). The limits of detection for calibrated VOC species are
below 100 ppt for the 1-s measurement, except for ethanol (423 ppt) and formic acid
(166 ppt). Additionally, the humidity dependence for a few VOC species in PTR-ToF-
MS (Yuan et al., 2017;Koss et al., 2018) were corrected using humidity-dependence
curves determined in the laboratory, as previously shown in Wu et al. (2020). To
quantify the ion signals without calibration, we determine the sensitivities based on the
kinetics of proton-transfer reactions of $H_3O^+$ with VOCs (Cappellin et al.,
2012;Sekimoto et al., 2017). The relationship between VOCs sensitivity and kinetic
rate constants for the same instrument has been reported in Wu et al. (2020) and He et
al. (2022). The corrected sensitivities as a function of kinetic rate constants for proton-
transfer reactions of $H_3O^+$ with VOCs during this campaign is shown in Fig. S5. The
fitted line is used to determine sensitivities of uncalibrated species, and the uncertainty
of the concentrations for uncalibrated species are determined to be around 50%.

## 2.3 Other VOC measurements

Whole air samples were collected using canisters after the dilution system for
determination of hydrocarbons emitted from various vehicles. All the canisters were
sent to the laboratory for analysis by an offline GC-MS/FID system, with a total 95
hydrocarbons calibrated by Photochemical Assessment Monitoring Stations (PAMS)
and TO-15 standard mixtures (Table S4). Due to the difference of sampling (e.g., times
and dilution ratios), we compared emission factors from PTR-ToF-MS and the offline
canister-GC-MS/FID, obtaining consistent results, except for gasoline vehicles with
China I (Fig. S6c).
An instrument based on Hantzsch reaction-absorption method was used to
measure formaldehyde (Zhu et al., 2020). Good agreement for formaldehyde between
PTR-ToF-MS and the Hantzsch instrument was obtained (Fig. S6a). An iodide-adduct
time-of-flight chemical ionization mass spectrometer (I⁻ ToF-CIMS, Aerodyne
Research, Inc.) (Wang et al., 2020c;Ye et al., 2021) was used to measure organic acids,



hydrogen cyanide (HCN), and isocyanic acid (HNCO) from vehicles (Li et al., 2021).
As shown in Fig. S6b, formic acid measured by PTR-ToF-MS and I⁻ ToF-CIMS showed
reasonable agreement.
**2.4 Emission factors and emission ratios calculation**

In this study, we determine emission factors of VOC species in two different

approaches: the mileage-based emission factors ($mg \cdot km^{-1}$) as the mass of these VOCs
exhausted per kilometer driving of vehicles, and the fuel-based emission factors
($mg \cdot kg_{fuel}^{-1}$) as the mass of VOCs per kilogram of fuel burned by the vehicles. In
addition, emission ratios of VOCs to combustion tracers (usually CO) are widely
applied in vehicle emissions in urban regions, as the result we determine emission
ratios to CO in $ppb \cdot ppm^{-1}$ as well. More details about the determination of emission
factors and emission ratios can be found in Sect. 1 in the Supplement.

The average emission factors for various types of vehicles are determined from

arithmetic means for different emission standards of vehicles. As for diesel vehicles,
the average emission factors are obtained from the arithmetic means of LDDT, MDDT,
HDDT, and bus. Besides, we also calculate emission factors and emission ratios from
weighted means based on the fractions of gasoline and diesel vehicles with different
emission standards in China (MEEPRC, 2019;Li et al., 2021) (See Sect. 1 in the
Supplement for details). In order to evaluate the uncertainties of obtained emission
factors, the average limit of detection for VOC species are used to estimate the limit
of detection for the determined emission factors (more details can be found in Sect. 2
in the Supplement).
**3. Results and discussions**
**3.1 Characteristics of the VOC emissions in the vehicles**

Time series of several aromatics and OVOC species measured by PTR-ToF-MS

for a selected gasoline vehicle associated with emission standard of China I and a LDDT
associated with China IV emission standard are shown in Fig. 1. Both tests started with
cold engines for the two vehicles. Benzene and toluene are typical aromatic species
exhausted by vehicles. As shown in Fig. 1a, high concentrations of benzene and toluene



exhausted by the gasoline vehicle were observed as the engine started. The
concentrations of the two species continued to increase until ~2 min after the engine
started, and then dropped rapidly before a minor increase during the acceleration
condition. These observations are similar to the previous results from PTR-MS
measurements in Drozd et al. (2016). Acetaldehyde and acetone are important OVOC
species emitted from vehicles. They show similar temporal variations as benzene and
toluene. However, concentrations of acetaldehyde and acetone were much lower than
the two aromatics after engine started. Compared to the concentrations at engine start-
up for the gasoline vehicle (the first cycle), concentrations of the VOCs are 3.0 to 40
times lower during the gasoline vehicle running at hot stabilized condition (the third
cycle). As shown in Fig. 1 for the diesel vehicle, enhanced emissions from cold start
are minor, which is different from the gasoline vehicle. The concentration of these
VOCs at engine start-up for the diesel vehicle are only 1.3 to 2.5 times higher than the
periods as the diesel vehicle running at hot stabilized condition. It indicates that the
impact of the engine start-up in diesel vehicles on emissions is much lower than
gasoline vehicles. It might be a combined effect of cold engine and operation
temperature of the after-treatment device. In contrast to the gasoline vehicle, we observe
higher concentrations of the two OVOC species than the two aromatics species from
the diesel vehicle. These higher OVOC concentrations in diesel vehicle exhausts are in
line with the observations of organic acids using the I- ToF-CIMS from the same
campaign (Li et al., 2021).
Based on the high time-resolution measurements of PTR-ToF-MS, we
determined emission factors of various VOC species from different vehicles. Fig. 2
shows the determined average mileage-based emission factors of benzene, toluene,
acetaldehyde, and acetone for various types of vehicles (also tabulated in the
Supplement table). In general, we observe a downward trend for emissions factors of
gasoline vehicles from China I to China VI emission standards for the four
representative VOC species. Emission factors of the four species for China VI vehicles
are 12 to 25 times lower than emissions for China I vehicles, indicating that newer
emission standards successfully reduced VOC emissions of gasoline vehicles. The


255 decline of emission factors for the four species with newer emission standards for diesel

256 vehicles are in the range of 1.1 to 7.4 times from China III to China V, compared to 4.5

257 to 5.4 times reduction from China III to China V for gasoline vehicles. Emission factors

258 of benzene and toluene from diesel vehicles are in the range of 0.8 to 7.4 mg·km$^{-1}$ and

259 0.3 to 5.8 mg·km$^{-1}$, which are comparable to emission factors from gasoline vehicles

260 with China IV to China VI emission standards. This is different from observations of

261 the two OVOC species (acetaldehyde and acetone), with much higher emission factors

262 from diesel vehicles (8.0 to 27.9 mg·km$^{-1}$ for acetaldehyde and 0.8 to 10.0 mg·km$^{-1}$ for

263 acetone) than almost all gasoline vehicles (a maximum of 3.9 mg·km$^{-1}$ for acetaldehyde

264 and a maximum of 3.2 mg·km$^{-1}$ for acetone). Higher emission factors from diesel

265 vehicles are also observed for many other common OVOC species, as shown in Fig. 3.

266 As the largest OVOCs emitted from gasoline vehicles ($4.6 \pm 5.1$ mg·km$^{-1}$), methanol is

267 found to be the only common OVOC species, with lower emission factors from diesel

268 vehicles than gasoline vehicles. The high emissions of OVOCs from diesel vehicles

269 may be related to combustion processes in diesel vehicles, with more excess air into

270 combustion cylinder resulting in higher oxygen contents and more oxidation processes

271 during fuel combustion (Pang et al., 2008;Qiao et al., 2012). Finally, the determined

272 emission factors of the four VOC species from LPG vehicles are much lower than both

273 gasoline and diesel vehicles.

## 274 3.2 Analysis VOCs of PTR-ToF-MS mass spectra

275   In addition to typical VOC species shown above, PTR-ToF-MS detected

276 abundant signals for a large number of ions. The determined average mileage-based

277 emission factors for all detected VOC species are shown as mass spectra in Fig. 4. VOC

278 species measured by PTR-ToF-MS were divided into groups according to chemical

279 formula, namely hydrocarbon species only containing C and H atoms ($C_xH_y$), OVOCs

280 ($C_xH_yO_z$), species containing nitrogen and/or sulfur atoms (N/S-containing), and some

281 other ions (others). We observe similar mass spectra of emission factors for gasoline

282 vehicles with different emission standards (Fig. S7). Highest emission factors from

283 gasoline vehicles (Fig. 4a) are detected as hydrocarbons, including $C_6$ to $C_{10}$ aromatics.





A few OVOC species, namely methanol, ethanol, formaldehyde, acetaldehyde and
acetone, are also observed as the largest emissions. In contrast to gasoline vehicles, the
largest emissions from diesel vehicles were attributed to a few low-molecular-weight
OVOC species, including formaldehyde, acetaldehyde, formic acid, and acetic acid,
followed by a large number of hydrocarbon species. Comparison between the mass
spectra of gasoline and diesel vehicle emissions suggest that emissions from diesel
vehicles are more evenly distributed among different VOC species, as reflected by 50
and 140 species contributing more than 1‰ of the total emissions for gasoline and diesel
vehicles, respectively. As shown in Fig. 3b, many hydrocarbon ions in the range of m/z
150-200 still account for significant fractions of emissions from diesel vehicles,
whereas only one species in this m/z range contribute more than 1‰ of emissions from
gasoline vehicles. These results demonstrate that diesel vehicles emit more heavier
hydrocarbons than those from gasoline vehicles, which is consistent with observations
in previous studies (Gentner et al., 2012;Erickson et al., 2014). It should be noted that
the signals of $C_{16}H_{22}O_4H$ (m/z=279) were higher during the tests based on determined
emission factors. However, we suspect that it may be emitted artifacts from the
sampling or dilution system as it mainly showed higher signals in the latter period of
each test when sampling materials absorb more heat from vehicle exhausts (Fig. S8),
and thus it is not included in Fig. 3 (details in the Sect. 2 in the Supplement).

The scatterplot of carbon oxidation states ($\overline{OS_C}$) as a function of carbon number

($n_C$) provides a framework for describing bulk chemical properties of organics (Kroll
et al., 2011). The details of $\overline{OS_C}$ calculation is included in Sect. 3 in the Supplement.
The results from gasoline and diesel vehicles are compared in Fig. 5 (LPG vehicles are
shown in Fig. S9). It is apparent that ions with carbon oxidation states between -2.0 to
0 comprise main emissions for each carbon number for both gasoline and diesel
vehicles. It is interesting to observe that averaged $\overline{OS_C}$ for $n_C$>6 increase as the carbon
number decrease for both gasoline and diesel vehicles, whereas the opposite trends are
observed for $n_C$<5. The averaged $\overline{OS_C}$ in diesel vehicles for $n_C$ between 1 and 5 are
significantly higher than those in gasoline vehicles, as the result of high emissions of
$C_2$ to $C_5$ low-molecular-weight OVOCs. Fig. 5c further shows that emission factors of





most VOC species from diesel vehicles were higher than gasoline vehicles, except a
number of species occupying in the right-bottom corner of the two-dimensional space.

The determined mass spectra of PTR-ToF-MS in terms of emission factor for

different types of vehicles can be used to explore the dependence of various VOC
emissions to different factors. Fig. 6a-b shows scatterplots of the average mileage-
based emission factors of VOCs between cold start and hot start for gasoline and diesel
vehicles, respectively. We observe strong correlation between emission factors from
cold start and hot start tests (R=0.99 and 0.92) and generally consistent ratios between
cold start and hot start for different types of VOC species for both gasoline and diesel
vehicles, indicating that variation behaviors are similar for different species and thus
chemical compositions of VOC emissions are comparable between different start
conditions. It is obvious that emission factors of VOCs during cold start are
significantly higher than those during hot start for gasoline vehicles (slope=0.40),
whereas similar emissions factors between cold start and hot start are derived for diesel
vehicles (slope=0.84). These results suggest that gasoline vehicles are more
significantly influenced by cold start, as the result of compositions in gasoline fuel are
more volatile than diesel fuel (US NRC, 1996). We further explore the effects of
emission standards to VOCs emission factors by comparing determined emission
factors between China I and China V for gasoline vehicle (Fig. 6c, also see China III
versus China V and China V versus China VI in Fig. S10) and between China III and
China V for LDDT (Fig. 6d, also see China III versus China V for MDDT and HDDT
in Fig. S10). Comparison of both gasoline and diesel vehicles demonstrate newer
emission standards successfully decreased VOC emissions. Based on the derived
slopes, we obtain VOCs emission factors reduced by a factor of 10 for gasoline
vehicles from China I to China V (a factor of 5 reduction from China III to China V
and a factor of 2.5 reduction for China V to China VI), and a factor of 2 reduction for
LDDT from China III to China V (a factor of 1.5 and 8 reduction for MDDT and
HDDT from China III to China V). The reduction ratio for gasoline vehicles from
China I to China V are generally similar for most VOC species, except that some
OVOC species with smaller reduction ratios. The reduction ratios for LDDT vehicles



from China III to China V show large variability for different species. The lowest
reduction ratios (a factor of ~2) are observed for the low-molecular weight OVOC
species associated with largest emissions, while the reduction ratios for hydrocarbons
and higher-molecular weight OVOCs are in the range of a factor of 10-100. These
results indicate the after-treatment device for diesel vehicles may effectively reduce
emissions of some heavier VOC species, though the after-treatment devices do not aim
for VOCs control.

### 3.3 Non-target analysis for comparison between gasoline and diesel vehicles

As shown in the previous section, the analysis of PTR-ToF-MS mass spectra
provide rich information on understanding the influences of VOC emissions from
vehicles. This detailed information provided by the PTR-ToF-MS also offer an
opportunity to systematically compare emissions between gasoline and diesel vehicles.
The scatterplot of the determined average emission factors of various VOC species
between gasoline and diesel vehicles is shown in Fig. 7. Large difference of VOC
compositions emitted from gasoline and diesel vehicles are observed, as indicated by
the low correlation of the data points (R=0.24). A limited number of VOC species,
including $C_6$-$C_{10}$ aromatics and some N/S-containing species (e.g. $C_7H_5N$) are
associated with higher emission factors from gasoline vehicles, whereas the obtained
emission factors of most VOC species emitted from diesel vehicles are higher,
especially most OVOC species. For example, formic acid is found to be one of the
most significant emission species in diesel vehicles, with emission factors three orders
of magnitude higher than that of gasoline vehicles. In addition, emission factors of
HCN from gasoline vehicles are similar to those from diesel vehicles. These results
are consistent with the measurements using the I$^-$ ToF-CIMS from the same campaign,
as shown in Li et al. (2021).
The scatterplot shown in Fig. 7 can also be expressed in terms of the determined
fuel-based emission factors between gasoline and diesel vehicles (Fig. S11). Generally,
similar variability is obtained except the determined slope of the data points, with





higher slopes determined from the scatterplot based on fuel-based emission factor
(0.19 versus 0.15). The difference between the slopes reflects the different average
mileage for the same weight of fuel between gasoline (9.7 km·kg$_{fuel}^{-1}$) and diesel
vehicles (7.1 km·kg$_{fuel}^{-1}$), as demonstrated for emission factors of $CO_2$ in Table S5.

From the comparison gasoline and diesel vehicles, we can also observe profound
differences in relative changes of emission factors for analogous compounds series. The
emission factors of $C_6$-$C_{10}$ aromatics are apparently higher for gasoline vehicles than
diesel vehicles, whereas emission factors for larger aromatics ($n_C$>11) from diesel
vehicles start to exceed gasoline vehicles. This interesting behavior is the result of
different variations of emission factors for gasoline and diesel vehicles as carbon
number increases. As shown in Fig. 8, emission factors of aromatics from gasoline
vehicles start to rapidly decrease at $n_C$=10 (a factor of 5 for each additional carbon for
$C_{10}$-$C_{15}$), while the emission factors of aromatic for diesel vehicles demonstrate a
relatively flat pattern between $C_6$ and $C_{15}$, only with significantly decrease for $n_C$>15.
Based on Fig. 8, we determine that emissions of aromatics with $n_C$≥10 in gasoline and
diesel vehicles are account for 14% and 63% of total aromatic emissions, again suggest
the importance of heavier aromatics in emissions from diesel vehicles. It also highlights
that quantification of these heavier species by PTR-ToF-MS may be important in
characterization of vehicular exhausts, especially diesel vehicles.

In addition to aromatics, the relative changes of emission factors for carbonyls
with carbon number are apparently different between gasoline and diesel vehicles (Fig.
7 and Fig. 8b). Emission factors of carbonyls tend to decrease as carbon number
increase for both gasoline and diesel vehicles. The decrease magnitudes are observed
to be comparable from $C_1$-$C_6$ carbonyls for gasoline (97.6%) and diesel vehicles
(97.4%). However, as $n_C$>6, the decrease of carbonyl emissions factors for diesel
vehicles become smaller, result in larger emissions factors than gasoline vehicles for
this range of carbon number.

The above discussions demonstrate that emission characteristics of aromatics and
OVOCs are significantly different between gasoline and diesel vehicles. As the result,
the ratios of VOC pairs can be identified to distinguish emissions of gasoline and diesel



vehicles. Fig. 9 shows the scatterplots of four representative VOCs (benzene, $C_{14}$
aromatics, formaldehyde, and acetaldehyde) versus toluene based on the determined
emission factors. The data points for each VOCs pair clearly show distinct separation
between gasoline vehicles and diesel vehicles, with apparently higher slopes for diesel
vehicles than gasoline vehicles, as the result of much larger emission factors of toluene
from gasoline vehicles and lower emission factors of the four representative VOCs
from diesel vehicles. The benzene/toluene ratio in gasoline and diesel vehicle are
determined as 0.48 and 1.24 $mg \cdot mg^{-1}$ (corresponding to 0.57 and 1.46 $ppb \cdot ppb^{-1}$ that
are more widely used in ambient studies). The difference of benzene/toluene ratio
between gasoline and diesel vehicles has been reported in previous studies, and our
results are generally consistent with these previous results (Chan et al., 2002;Barletta
et al., 2005;Qiao et al., 2012;Kumar et al., 2020). Compared to benzene/toluene ratio,
the difference of $C_{14}$ aromatics/toluene ratio between gasoline and diesel vehicles are
more substantial (a factor of 3800). The remarkable larger emission factors of $C_{14}$
aromatics from diesel vehicles suggest that diesel vehicles can be a significant or even
predominated source for higher molecular aromatics. The enormous difference of $C_{14}$
aromatics/toluene ratio (and also other higher aromatics/toluene) between gasoline
and diesel vehicles indicate these ratios could potentially provide good indicators for
separation of gasoline and diesel vehicles in ambient or tunnel studies. Similar
discrepancies are observed for formaldehyde/toluene and acetaldehyde/toluene ratios
between gasoline and diesel vehicles. These ratios may not be able to be used as
indicators for distinguish gasoline and diesel vehicles in ambient studies, since
secondary sources may complicate the observed ratios in ambient air. However, these
results strongly suggest that diesel vehicles can be important in emissions of these
OVOC species, though the number of diesel vehicles are smaller than gasoline
vehicles in many countries, e.g. China and U.S (Wallington et al., 2013;Yao et al.,
2015;Huang et al., 2021).
**3.4  OVOC fractions in VOC emissions**
Emission factors of various VOC species measured by PTR-ToF-MS from





different vehicles are summarized in Fig. 10. As shown in Fig. 10a, the determined
average mileage-based emission factors of total VOC ions from diesel vehicles were
much higher than gasoline and LPG vehicles. Fig. 10b-d quantified the proportions of
different categories of ions measured by PTR-ToF-MS. The determined average
mileage-based emission factors of $C_xH_y$ accounted for the largest fraction in gasoline
vehicles (84% ± 5.9%), and lower fractions in diesel (47% ± 16%) and LPG vehicles
(32% ± 0.7%). OVOCs account for larger fractions in diesel (49% ± 16%) and LPG
vehicles (58% ± 3.7%), while they only account for 13% ± 6.1% of emissions from
gasoline vehicles. The fractions of different OVOC groups generally demonstrate a
downward trend from $C_xH_yO_1$ to $C_xH_yO_{\geq 3}$, and OVOCs with more than two oxygen
atoms only occupy small percentages (0-7%) in vehicle exhausts, indicating low
emissions of these species.

Combined with measurements of other VOCs from canisters measured by GC-

MS/FID, the fractions of OVOCs in total VOC emissions can determined for different
vehicles (details in Sect. 4 in the Supplement) (Fig. 11). OVOCs account for 7.7% ±
6.2% of total VOC emissions for gasoline vehicles. The OVOC fractions for gasoline
vehicles are generally comparable for different emission standards and cold/hot start,
except somewhat higher fractions for China VI from hot start (Fig. S12). The OVOC
fractions obtained in this study for gasoline vehicles are generally consistent with
previous results (Cao et al., 2016;Wang et al., 2020b) (Fig. 11). Among these studies,
the OVOC fractions determined for gasoline with 10% ethanol (E10) (Roy et al., 2016)
(22% ± 11%) are apparently higher. The fractions of OVOCs in total VOC emissions
for diesel vehicles are 77% ± 15%, 68% ± 15%, 73% ± 14%, and 40% ± 10% for LDDT,
MDDT, HDDT, and bus, respectively. The variations of OVOC fractions with emission
standards are observed to be mixed among different types of diesel vehicles (Fig. S12).
The OVOC fractions from diesel vehicles are obviously higher than those in gasoline
vehicles, indicating the importance of OVOCs in VOC emissions for diesel vehicles.
Compared to previous studies (Tsai et al., 2012;Qiao et al., 2012;Cao et al., 2016;Mo
et al., 2016), determined OVOC fractions for diesel vehicles in this study are higher. If
only considering carbonyls among various types of OVOCs measured by PTR-ToF-MS,





the OVOC fractions determined in this study are more comparable with previous studies (Fig. 11), since most previous studies only detected carbonyls among various types of OVOCs. Finally, we determine that OVOCs account for 41% ± 8.6% of total VOC emissions for LPG vehicles, which is also higher than in one previous study (Wang et al., 2020b) with only carbonyls and a few esters/alcohols included. These results stress that the large number of OVOCs measured by PTR-ToF-MS are important in characterization of VOC emissions from vehicles. It should be noted that the OVOC fractions obtained here only reflect exhaust emissions. Evaporative emissions may be associated with different fractions of various VOC groups, which may be more related to fuel compositions (Rubin et al., 2006;Huang et al., 2021).

## 4. Conclusions

In this work, we conducted a chassis dynamometer study to measure VOC emissions from gasoline, diesel, and LPG vehicles using PTR-ToF-MS along with other offline and online measurement techniques. Using this dataset, we provide emission factors of many VOCs from these three different types of vehicles associated with various emission standards in China. Our results show that emission factors of VOCs generally decrease with the increased stringency of emission standards for gasoline vehicles, whereas variations of emission factors for diesel vehicles with emission standards are more diverse. Mass spectra analysis of PTR-ToF-MS suggest that cold start significantly influence VOCs emission of gasoline vehicles, while the influences are smaller for diesel vehicles.

We observe large differences of VOC emissions between gasoline and diesel vehicles based on PTR-ToF-MS measurements. Emission factors of most VOC species from diesel vehicles were higher than gasoline vehicles, especially for most OVOCs and heavier aromatics. The substantial larger emission factors of some OVOCs emission factors for diesel vehicles indicate potentially dominant emissions of these species from diesel vehicles among vehicular emissions. Our results suggest that VOC pairs (e.g. $C_{14}$ aromatics/toluene ratio) could potentially provide good indicators for distinguishing emissions between gasoline and diesel vehicles.



Based on measurements of PTR-ToF-MS, $C_xH_y$ ions account for the largest
fraction in gasoline vehicles (84% ± 5.9%), whereas OVOC ions are the largest
contributor in the mass spectra of emissions from diesel (49% ± 16%) and LPG vehicles
(58% ± 3.7%). In the end, the fractions of OVOCs in total VOC emissions are
determined by combining hydrocarbons measurements from canister results and online
measurements of PTR-ToF-MS. We show that OVOCs contribute 7.7% ± 6.2% of
gasoline vehicles of the total VOC emissions, while the fractions are significantly
higher for diesel vehicles (40-77%), highlighting the importance to detect these OVOC
species in diesel emissions.
This study shows significant contributions of OVOCs in VOC emissions from
various vehicles, especially diesel vehicles. As a consequence, vehicular emissions may
account for considerable proportions for primary emissions of these OVOCs in urban
regions. Emissions of many OVOC species are currently not fully represented in
emission inventories of VOCs, which may in turn affect the prediction ability of air
quality models in urban regions. In this study, OVOC species are mainly quantified
from PTR-ToF-MS measurements by taking into account all signals in the mass spectra,
which stress that the large number of OVOC species measured by PTR-ToF-MS are
important in characterization of VOC emissions from vehicles.
**Data availability**
Data are available from the authors upon request.
**Author contribution**
BY designed the research. ZBY, JYZ, BY, QES organized vehicle test
measurements. SHW, CHW, CMW, TGL, JPQ, QES, and MMZ contributed to data
collection. SHW performed the data analysis, with contributions from TGL, XJH, YBH,
XBL, and QES. SHW and BY prepared the manuscript with contributions from other
authors. All the authors reviewed the manuscript.
**Competing interests**
The authors declare that they have no known competing financial interests or
personal relationships that could have appeared to influence the work reported in this



paper.

## Acknowledgement

This work was supported by the National Key R&D Plan of China (grant No.
2019YFE0106300, 2018YFC0213904), the National Natural Science Foundation of
China (grant No. 41877302, 42121004), Guangdong Natural Science Funds for
Distinguished Young Scholar (grant No. 2018B030306037), and Guangdong
Innovative and Entrepreneurial Research Team Program (grant No. 2016ZT06N263).
This work was also supported by Special Fund Project for Science and Technology
Innovation Strategy of Guangdong Province (Grant No.2019B121205004). TK and
MG were supported by OEAD grant CN 05/2020.



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



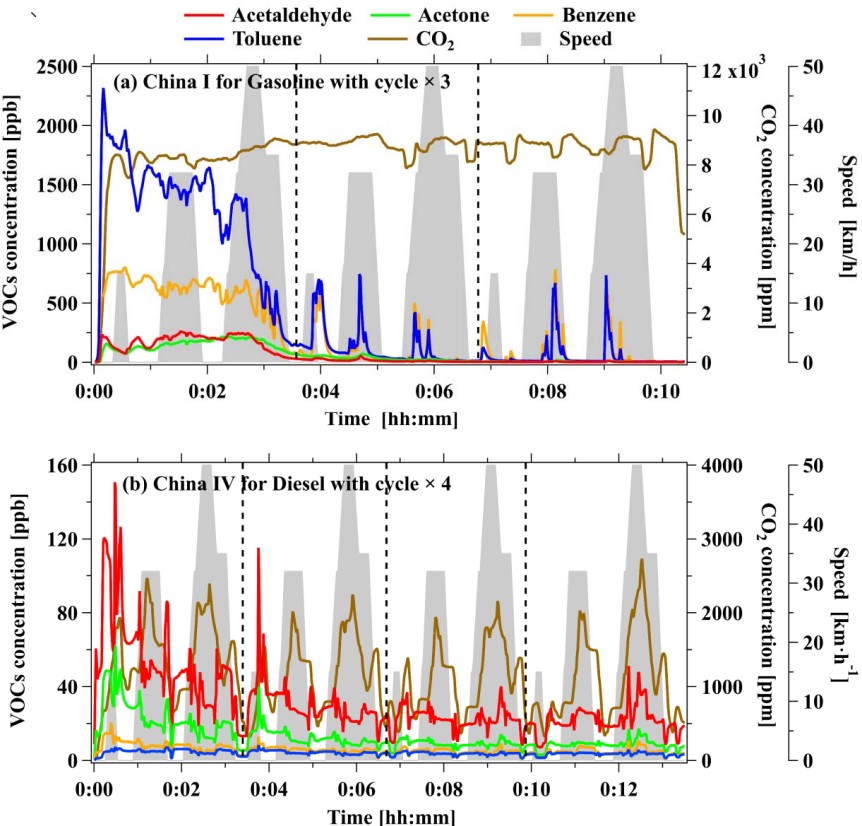

**Figure 1.** Real-time concentrations of acetaldehyde, acetone, benzene, toluene, and $CO_2$ for (a) a gasoline vehicle with emission standard of China I and (b) a light-duty diesel vehicle with emission standard of China IV. The two vehicles were both cold started. The gray shadows represent the speed of the vehicles on the chassis dynamometer.

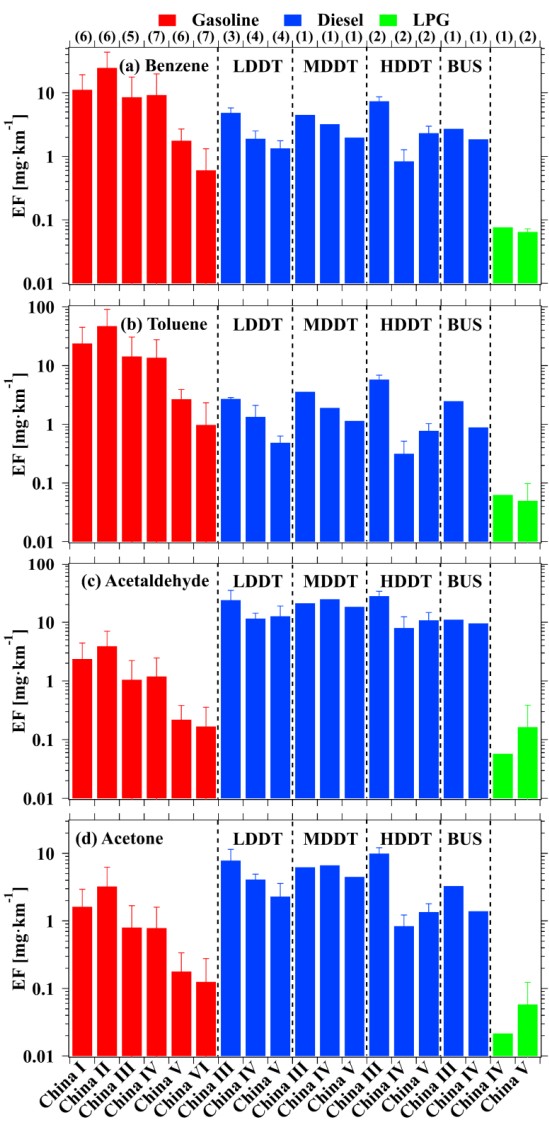

**Figure 2.** The determined average mileage-based emission factors (mg·km⁻¹) for (a) benzene, (b) toluene, (c) acetaldehyde, and (d) acetone for vehicles with different emission standards. The numbers above the top axis represent the number of all experiments (including multiple measurements for individual test vehicle) for each emission standard. Error bars represent standard deviations of emission factors for the specific emission standard.





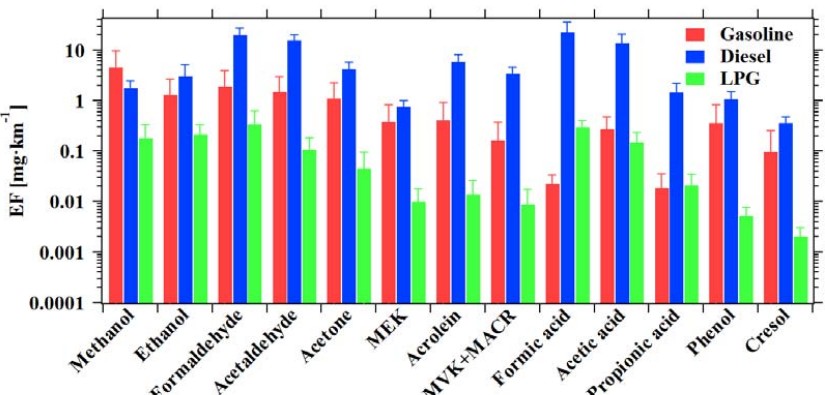

803

**Figure 3.** The determined emission factors of representative OVOC species from different types of vehicles. Error bars represent standard deviations of the emission factors for the VOCs.

807

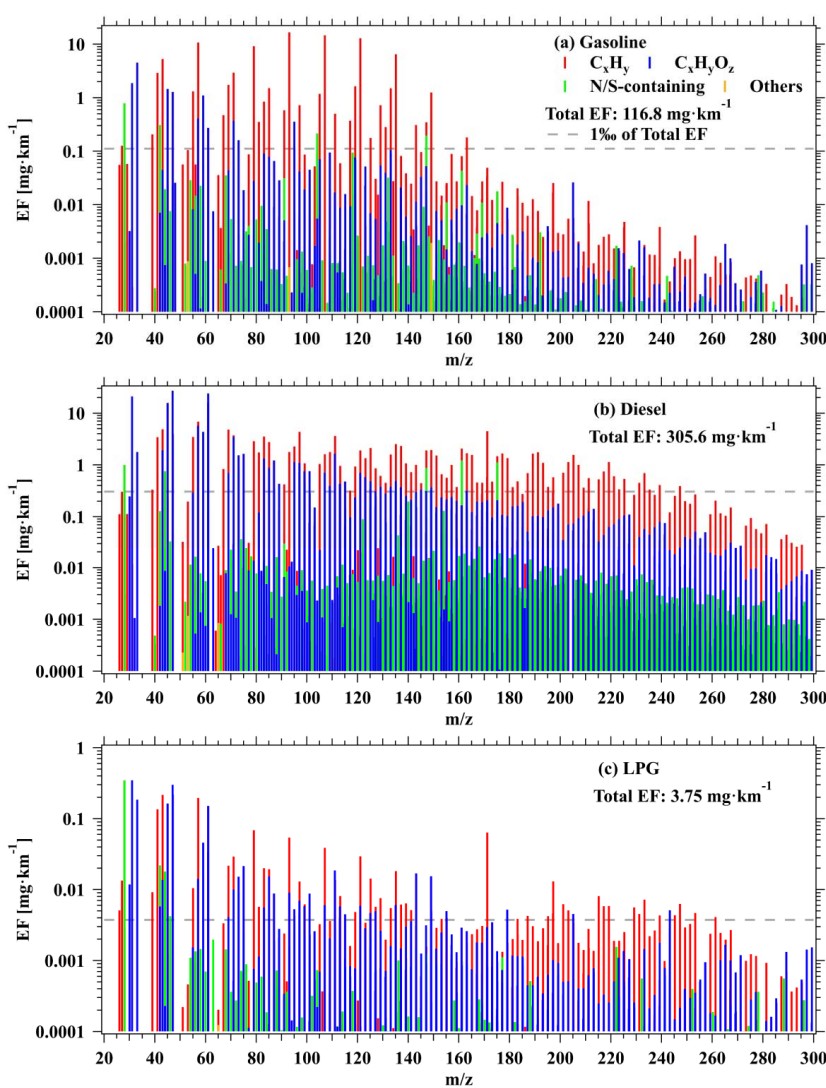

**Figure 4.** The determined average mileage-based emission factors of VOC species measured by PTR-ToF-MS from (a) gasoline, (b) diesel, and (c) LPG vehicles. The gray dashed lines represent 1‰ of total VOCs emission factors.

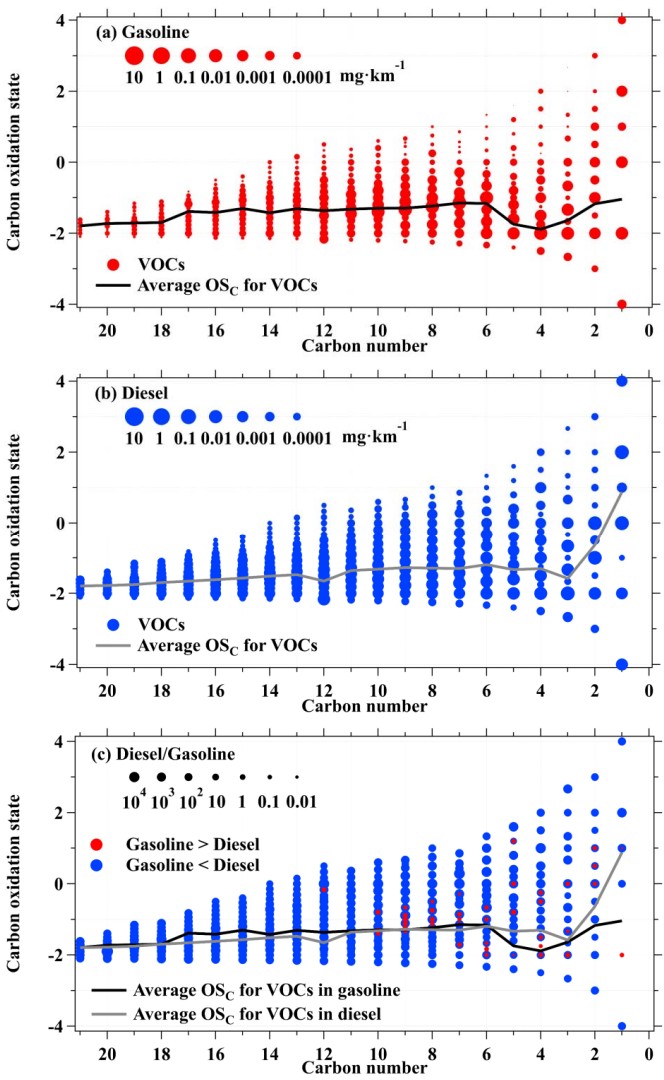

**Figure 5.** The two-dimensional space of $\overline{OS_C} - n_C$ with data points sized coded using emission factors of VOC species from (a) gasoline and (b) diesel vehicles, and (c) the ratio of emission factors of diesel vehicle relative to gasoline vehicle. The black and gray lines are the average $\overline{OS_C}$ of each carbon number for VOC species in gasoline and diesel vehicles, respectively.

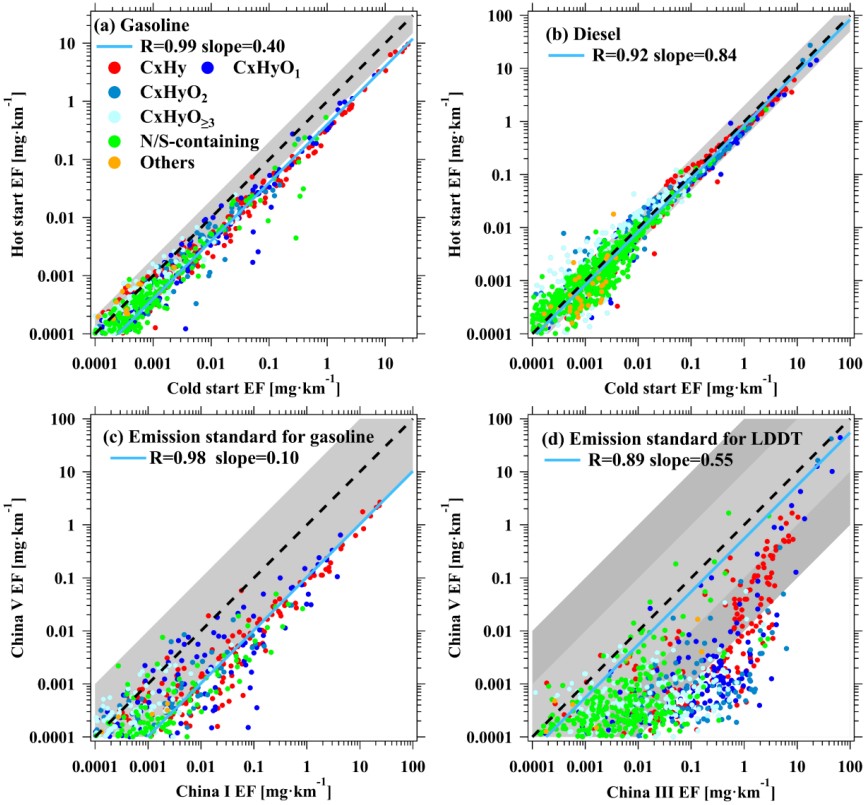

**Figure 6.** Scatterplots of VOCs emission factors between cold start and hot start for gasoline (a) and diesel vehicles (b). Scatterplots of VOCs emission factors between China I and China V emission standard for gasoline vehicles (c) and between China III and China V emission standard for diesel vehicles (d). Each data point indicates a VOC species measured by PTR-ToF-MS. The blue lines are the fitted results for all data points. The black dashed lines represent 1:1 ratio, and the shaded areas represent ratios of a factor of 2 in (a) and (b), and a factor of 10 and 100 in (c) and (d).



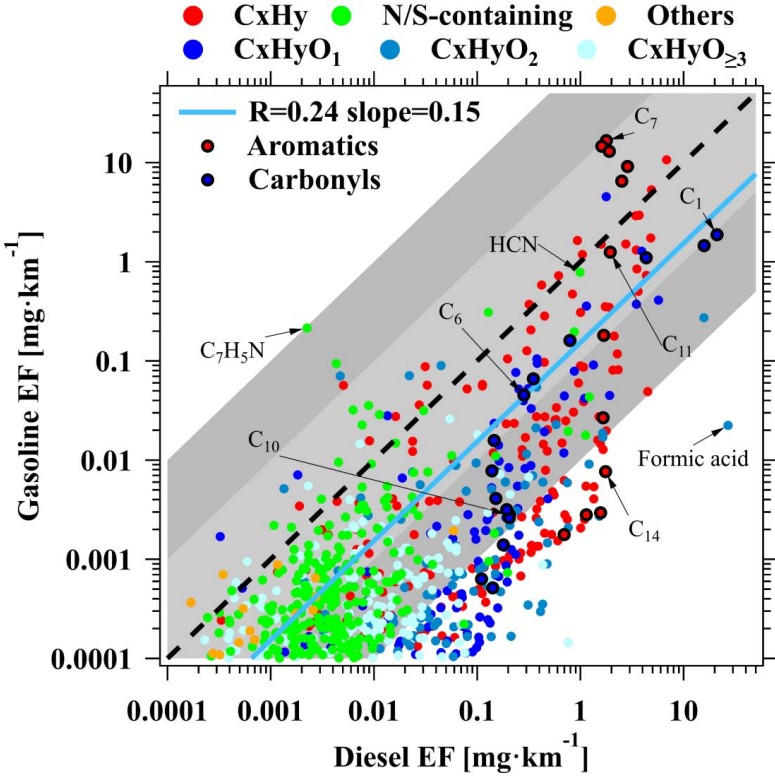

829

**Figure 7.** Scatterplot of VOCs emission factors between gasoline and diesel vehicles.

Each data point indicates a VOC species measured by PTR-ToF-MS. The blue line is

the fitted result for all data points. The black line represents 1:1 ratio, and the shaded

areas represent ratios of a factor of 10 and 100.

834



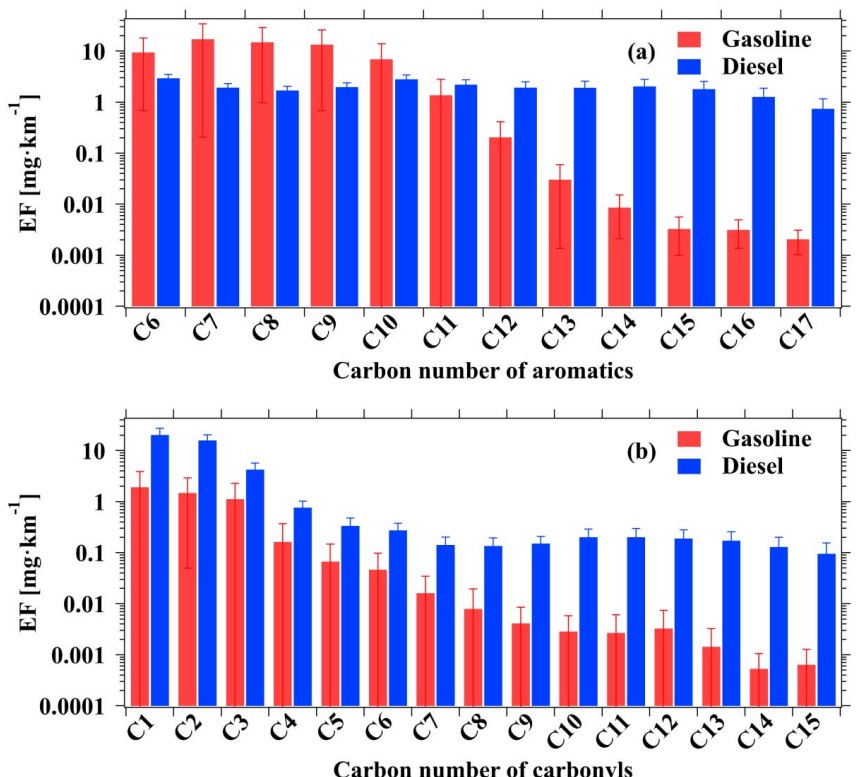

835

**Figure 8.** The determined emission factors of (a) aromatics and (b) carbonyls for each carbon number from gasoline and diesel vehicles. Error bars represent standard deviations of the emission factors for the VOCs of different carbon number.

839

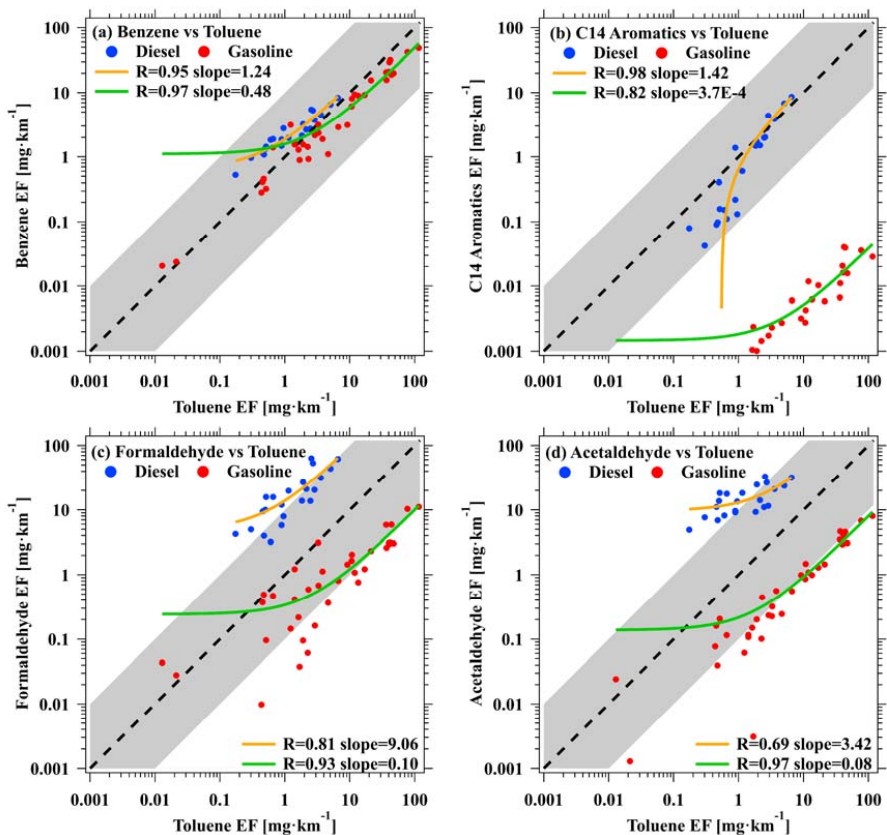

**Figure 9.** Scatterplots of the determined mileage-based emission factors of (a) benzene versus toluene, (b) $C_{14}$ aromatics versus toluene, (c) formaldehyde versus toluene, and (d) acetaldehyde versus toluene for gasoline and diesel vehicles. Each data point represents each test vehicle in this study. The green and orange lines are the fitted results for gasoline and diesel vehicle. The black line represents 1:1 ratio, and the shaded areas represent ratio of a factor of 10.



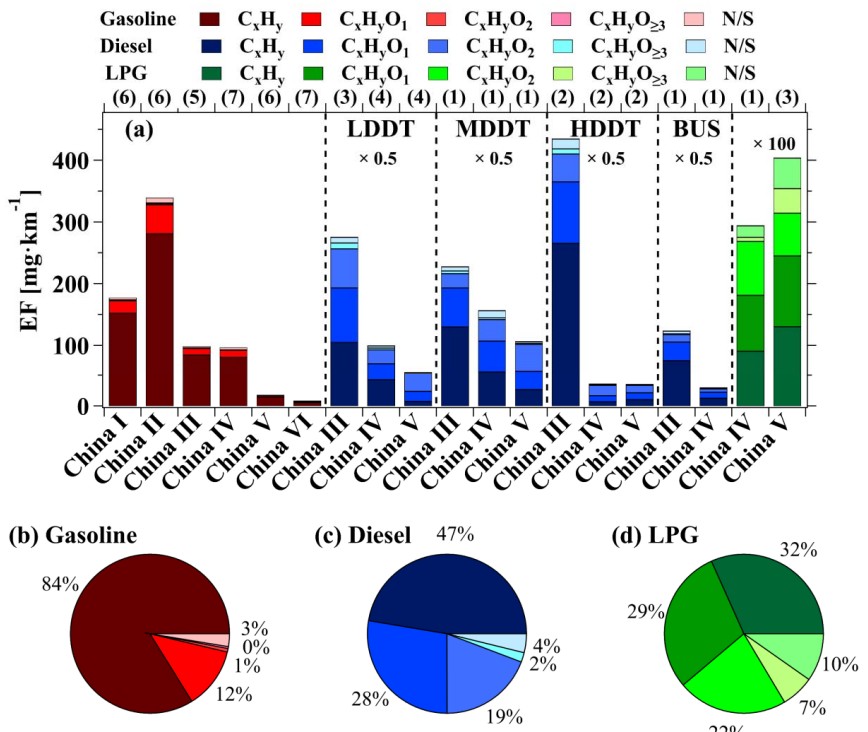

**Figure 10.** (a) The determined average emission factors for different emission standard from gasoline, diesel (×0.5), and LPG (×100) vehicles measured by PTR-ToF-MS. The different ion categories are discussed in the manuscript. Fractions of the determined average emission factors of VOCs ions in different ion categories from (b) gasoline, (c) diesel, and (d) LPG vehicles. The numbers above the top axis represent the number of all experiments (including multiple measurements for individual test vehicle) for each emission standard.

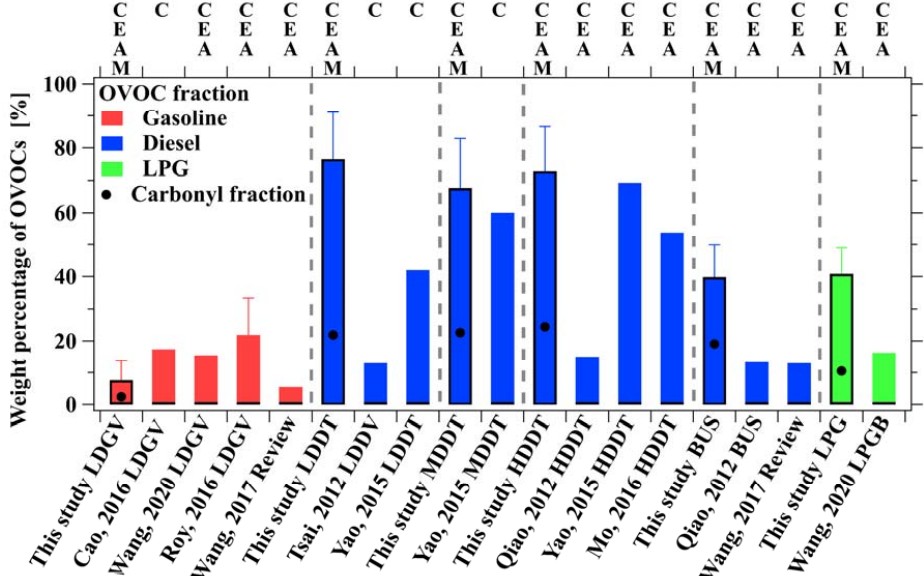

**Figure 11.** Comparison of OVOCs fractions determined in this study and those in previous studies. Error bars represent the standard deviations of the weight percentage of OVOCs. The C, E, A, M above the top axis represent the four groups of OVOCs measured in this study or previous studies, including Carbonyl: C, Ester/Ether: E, Alcohol: A, Multiple-functional: M.