# Peer review of "Oxygenated VOCs as significant but varied contributors"

_Atmospheric Chemistry and Physics, 2022_

## Author Comment (AC1)

**Response to Reviewers**

**Reviewer #1**

*Overview*

*This manuscript characterizes gaseous emissions from a number of vehicles meeting a wide range of Chinese emissions standards, to include gasoline, diesel, and liquified petroleum gas (LPG), as measured using a chassis dynamometer setup. Measurements are primarily presented for those using a PTR-ToF-MS, and included canister sampling with GC-MS/FID analysis and a few species by Iodide CIMS, along with common measurements ($CO_2$, etc.) using a portable emissions measurement system. Oxygenated VOC's (OVOC) are indicated to be molecules with less than 18 carbons.*

*This work shows the strong influence of OVOC in diesel exhaust (>50% by mass) compared to a much smaller influence in gasoline vehicles (~15%). Clear differences between cold-start and hot-start emissions are also observed, notably they are much more significant for gasoline vehicles than for diesel vehicles, and aromatics and OVOC had similar temporal profiles. Some ratios of emissions (e.g. toluene to larger aromatics) are unique between gasoline and diesel vehicles, and are suggested as potentially useful for emissions attribution.*

*Overall the work as presented is quite thorough, and the intended goals of the work are clearly made. The insights from the work are a good contribution to the field. There are a few details that should be addressed, however prior to suitability for publication, notably quality control.*

Reply: We would like to thank the reviewer for the insightful comments, which helped us tremendously in improving the quality of our work. Please find the response to individual comments below.

*General Comments*

*1. It is concerning that the agreement between canister with GC-MS/FID and PTR-ToF measurements for toluene are so disparate in more than 20% of the tested vehicles, as shown in Figure S6c. These discrepancies are essentially ignored in the manuscript.*

*How can canister measurements be near-zero while PTR-ToF measurements are 250 mg/km, and vice versa? Perhaps this might cold occur for more exotic species, but for toluene I would expect agreement at least within a factor of 2 in all cases, as it is a high volatility species that is easily ionized in PTR and observed with GC-MS/FID. Perhaps in some cases one or the other measurement was not made and simply reported as zero? This issue should be clarified. Furthermore, agreement between generally accepted canister measurements and PTR-ToF measurements must be reported for a wider variety of species, to include oxygenated species and larger aromatics.*

Reply: We thank the reviewer for the comment. We also found this issue when we analyzed data for preparing the original manuscript, but we did not pay enough attention to it. We re-checked all the data and found a mistake in alignment of data points between offline canister-GC-MS/FID and PTR-ToF-MS for several gasoline vehicles and diesel vehicles. We have modified the corresponding data points, and added comparison of $C_8$ aromatics between two measurements (Fig. S6), obtaining generally consistent results, considering large variations of VOC emissions for driving conditions and the difficulty to control the fill time for canisters. We also revised related figures (Fig. 12 and Fig. S12) and description in the manuscript on the fractions of OVOCs in total VOC emissions in various types of vehicles. These modifications do not change any conclusion in the manuscript.

The sentence in the Section 2.3 (line 194-197) is modified to:

**We compared emission factors from PTR-ToF-MS and the offline canister-GC-MS/FID (Fig. S6c-d), obtaining generally consistent results, considering the large variation of VOC emissions for driving conditions and the difficulty to control the fill time for canisters.**

[Figure]

**Figure S6. (a) Time series of formaldehyde measured by PTR-ToF-MS and the**

**Hantzsch instrument. (b) Scatterplot of the concentration of formic acid between**

**PTR-ToF-MS and the CIMS. Scatterplot of the emission factor of (c) toluene and**

**(d)$C_8$ aromatics calculated by the data detected by PTR-ToF-MS and Canister-**

**GC-MS/FID. The black dashed lines represent 1:1 ratio, and the shaded areas**

**represent ratios of a factor of 2 and 10 in (c) and (d).**

[Figure]

**Figure 12. Comparison of OVOCs fractions determined in this study and those in previous studies. Error bars represent the standard deviations of the weight percentage of OVOCs. The C, E, A, M above the top axis represent the four groups of OVOCs measured in this study or previous studies, including Carbonyl: C, Ester/Ether: E, Alcohol: A, Multiple-functional: M.**

[Figure]

**Figure S12. (a) Average OVOC fractions for vehicles with different emission standards, and some difference between (b) cold start and (c) hot start. Error bars represent the standard deviations of the fraction of OVOCs.**

*2. The mileage of the vehicles tested is quite variable, are there any correlations in your data with mileage, are these different for gasoline vs. diesel?*

Reply: We thank the reviewer for the comment. The mileage of the vehicles is one of determining factors of emissions from the vehicles. We also found that the emission factors for the representative VOC species in China II gasoline vehicles are higher than other emission standards, which may be explained by the higher mileage of them than other vehicles (Fig. 3). Strong positive correlations between emission factors and mileage are obviously for both gasoline and diesel vehicles. We added some description in the Section 3.1 and Fig. 3 with scatterplot of the emission factor of (a)-(b) toluene and (c)-(d) acetone during the hot start based on the odometer for each gasoline and diesel vehicle.

The sentences in the Section 3.1(line 258-270) are modified to:

**In general, we observe a downward trend for emissions factors of gasoline vehicles from China I to China VI emission standards for the four representative VOC species. This is consistent with the results in previous studies with lower emissions for newer emission standards (Wang et al., 2017;Sha et al., 2021). In addition, the dependence of VOCs emission versus emission standard may also be attributed to the history of vehicle usage, i.e., the mileage traveled by the vehicles, as lower mileages of vehicles are usually associated with vehicle with newer emission standards. As shown in Fig. 3, we observe strong positive relationship between toluene emission factors and vehicle odometers for both gasoline and diesel vehicles, indicating the mileages of vehicles can significantly affect VOCs emission factors for vehicles tested in this study. Intestinally, the emission factors of the representative VOC species are highest for China II gasoline vehicles rather than China I vehicles, coincidence with largest mileage of the test vehicles.**

[Figure]

**Figure 3. Scatterplot of the emission factor of toluene in (a) gasoline and (b) diesel vehicles, and acetone in (c) gasoline and (d) diesel vehicles during the hot start based on the odometer for each vehicle.**

*3. Was any analysis of the fuels done? To make clear sense of the emissions, the compositions of each fuel type, in terms of saturates (linear and cyclic), aromatics (BTEX and others), and oxygenates should be given. This is especially important for the diesel fuel, which can vary significantly in terms of aromatic content. Were the fuels summer or winter blends? The results presented have much narrower significance without clearer information on the fuel composition. Did the gasoline fuel have any ethanol content, as might be expected for gasoline in China after 2017? Ethanol content will have significant effects on small OVOC emissions. The discussion beginning on line 377 is well explained by the difference in aromatic content of the two fuels.*

Reply: Thanks for the reviewer's advice. Fuel composition is one of determining factor for VOCs emissions from vehicles (Gentner et al., 2017). However, the compositions of fuel were not measured during the tests, as most of the test vehicles are mainly from the local automobile quality supervision test center in this study. A fraction of vehicles is from a car rental company, with full tank of fuel before the test. In response to the reviewer's comment, we conducted some literature review and added a discussion in Section 1 in the Supplement to provide some information about chemical compositions of gasoline and diesel fuel in China, and added some description of difference in aromatic content of the two fuels in the Section 3.3. Furthermore, gasoline and diesel fuel are summer blends, and the gasoline fuel does not content ethanol in this study.

The sentence in the Section 2.1 (line 126-127) is modified to:

**The detailed information for test vehicles is summarized in Sect. 1 in the Supplement, Table S2 and Table S3.**

The Section 1 in the Supplement is modified to:

**Fuel composition is one of determining factor for VOCs emissions from vehicles (Gentner et al., 2017). The gasoline fuel used in China is mainly comprised of $C_4$-$C_7$ hydrocarbons. The chemical compositions of gasoline fuel are alkanes (55%-62%), alkenes (12%-17%), aromatics (27%-32%), and methyl tert-butyl ether (MTBE, 1%-4%) (Tang et al., 2015;Sun et al., 2021;Qi et al., 2021;Huang et al., 2022). Heavy hydrocarbons, namely $C_8$-$C_{10}$ alkanes and aromatics, contributed most in diesel fuel. The chemical compositions of diesel are alkanes (70%-79%), alkenes (1%-7%), and aromatics (21%-25%) (Wang et al., 2015;Yue et al., 2015;Hou and Jiang, 2018;Liu and Zhang, 2015). Gasoline and diesel fuel are summer blends, and the gasoline fuel does not content ethanol in this study.**

The sentences in the Section 3.3(line 416-420) are modified to:

**This interesting behavior is the result of different variations of emission factors for gasoline and diesel vehicles as carbon number increases. This may be attributed to the differences of chemical compositions of gasoline and diesel fuel, such as higher fractions of polycyclic aromatic hydrocarbons (PAHs) in the diesel fuel (Yue et al., 2015;Gentner et al., 2017).**

*4. When considering the usefulness of ratios between emitted species as diagnostic for*

*diesel vs. gasoline species, you should also consider their atmospheric lifetimes for*

*oxidation.*

Reply: We thank the reviewer for the suggestion. It is necessary to consider the atmospheric lifetime of $C_{14}$ aromatics and toluene for oxidation when used the $C_{14}$

aromatics/toluene ratio as diagnostic for diesel versus gasoline vehicles. Here, we consider the change of $C_{14}$ aromatics/toluene ratio with the OH reaction in the atmosphere (de Gouw et al., 2005) (Figure R1):

$$\frac{[C_{14} \text{ aromatics}]}{[Toluene]} = ER \times exp[-(k_{C14} - k_{Toluene})[OH] \times t] \quad (6)$$

Where $[C_{14} \text{ aromatics}]$ and $[Toluene]$ are the concentrations of $C_{14}$ aromatics and toluene, respectively. $ER$ is the emission ratio of $C_{14}$ aromatics versus toluene (1.42

in diesel vehicles and 3.7E-4 in gasoline vehicles). $[OH]$ is the concentration of OH

radicals (mole·cm$^{-3}$). $k$ is the rate constant of the OH reaction with toluene (5.63 ×10$^{-}$

$^{12}$ cm$^3$·mole$^{-1}$·s$^{-1}$) (Atkinson and Arey, 2003) and $C_{14}$ aromatics, respectively. $t$ is the photochemical age. Here, an averaged OH concentration in the PRD, China with

$1.5×10^6$ mole·cm$^{-3}$ is used (Wang et al., 2020;Tan et al., 2019). Due to the rate constant for $C_{14}$ aromatics did not report in previous study, we used the averaged rate constant for $C_{12}$ aromatics (hexamethylbenzene) to estimate the reaction rate (1.33 ×10$^{-10}$

cm$^3$·mole$^{-1}$·s$^{-1}$) (Alarcon et al., 2015;Berndt and Böge, 2001), which may be a little lower than the real value of $C_{14}$ aromatics rate constant.

Based on the Equation (6), the $C_{14}$ aromatics/toluene ratio emitted from diesel vehicles will be higher than emission ratio of gasoline vehicles for photochemical reactions shorter than 12 h. Therefore, the $C_{14}$ aromatics/toluene ratio could be applied to the ambient measurements in urban or downwind regions, especially for roadside measurements or tunnel study to distinguish the emission of diesel and gasoline vehicles. Therefore, we conclude that the $C_{14}$ aromatics/toluene ratio should be applied for distinguishing emissions of gasoline and diesel vehicles in ambient measurements of urban or downwind regions, especially for roadside measurements or tunnel study to distinguish the emission of diesel and gasoline vehicles.

The sentence in the Section 3.3 (line 457-461) is modified to:

**The enormous difference of $C_{14}$ aromatics/toluene ratio (and also other higher aromatics/toluene) between gasoline and diesel vehicles indicate these ratios could potentially provide good indicators for separation of gasoline and diesel vehicles in ambient or tunnel studies (see discussion in Sect. 5 in the Supplement for details about the feasibility of the ratio using in ambient air).**

We added a discussion in Section 5 in the Supplement and Fig. R1 to provide the feasibility of the $C_{14}$ aromatics/toluene ratio used as diagnostic parameter for diesel versus gasoline vehicles. The Section 5 in the Supplement is modified to:

**It is necessary to consider the atmospheric lifetime of $C_{14}$ aromatics and toluene for oxidation when used the $C_{14}$ aromatics/toluene ratio as diagnostic for diesel versus gasoline vehicles. Here, we consider the change of $C_{14}$ aromatics/toluene ratio with the OH reaction in the atmosphere (de Gouw et al., 2005):**

$$\frac{[C_{14}\ \mathbf{aromatics}]}{[\mathbf{Toluene}]} = ER \times exp[-(k_{C14} - k_{Toluene})[OH] \times t] \quad (6)$$

**Where $[C_{14}\ \mathbf{aromatics}]$ and $[\mathbf{Toluene}]$ are the concentrations of $C_{14}$ aromatics and toluene, respectively. $ER$ is the emission ratio of $C_{14}$ aromatics versus toluene (1.42 in diesel vehicles and 3.7E-4 in gasoline vehicles). $[OH]$ is the concentration of OH radicals (mole·cm$^{-3}$). $k$ is the rate constant of the OH reaction with toluene (5.63 ×10$^{-12}$ cm$^3$·mole$^{-1}$·s$^{-1}$) (Atkinson and Arey, 2003) and $C_{14}$ aromatics, respectively. $t$ is the photochemical age. Here, an averaged OH concentration in the PRD, China with 1.5×10$^6$ mole·cm$^{-3}$ is used (Wang et al., 2020;Tan et al., 2019). Due to the rate constant for $C_{14}$ aromatics did not report in previous study, we used the averaged rate constant for $C_{12}$ aromatics (hexamethylbenzene) to estimate the reaction rate (1.33 ×10$^{-10}$ cm$^3$·mole$^{-1}$·s$^{-1}$) (Alarcon et al., 2015;Berndt and Böge, 2001), which may be a little lower than the real value of $C_{14}$ aromatics rate constant.**

**Based on the Equation (6), the $C_{14}$ aromatics/toluene ratio emitted from**

**diesel vehicles will be higher than emission ratio of gasoline vehicles for photochemical reactions shorter than 12 h. Therefore, the $C_{14}$ aromatics/toluene ratio could be applied to the ambient measurements in urban or downwind regions, especially for roadside measurements or tunnel study to distinguish the emission of diesel and gasoline vehicles. Therefore, we conclude that the $C_{14}$ aromatics/toluene ratio should be applied for distinguishing emissions of gasoline and diesel vehicles in ambient measurements of urban or downwind regions, especially for roadside measurements or tunnel study to distinguish the emission of diesel and gasoline vehicles.**

[Figure]

Figure R1. The volume mixing ratios of $C_{14}$ aromatics/toluene in diesel vehicles and gasoline vehicles versus the photochemical age. The black line represents emission ratio of $C_{14}$ aromatics versus toluene in gasoline vehicles.

*Specific Comments*

*1. Figure 9. The fits to your data are poorly presented this way, either for predicting the values through the whole range or for giving physical insight. Perhaps you should make the axes linear rather than logarithmic. At the very least, you should explain that the strange curves to these linear fits in log-log space are due to the y-intercept, or perhaps only plot these fits in the region where they appear linear (where the intercept is small compared to the fit value) and note that you plot only in the region of reasonable fit.*

Reply: We thank the reviewer for the comment. Many previous studies have used the logarithmic axes in plots to better demonstrating the large variability in emission factors for different organic compounds (Gentner et al., 2013;Gentner et al., 2017;Zhao et al., 2016). We also tried to change the plot with linear axes (Fig. R2), with much poor performance for the data points associated with lower emission factor. We added some description in the caption of Fig. 10 in the revised manuscript (Fig. 9 in the original manuscript) about the counterintuitive non-linear curves for a line with non-zero y- intercept in log-log space.

**The green and orange line are the fits to gasoline and diesel points in each plot.**

**Note that these linear fits are shown in curves in log-log space as the result of non-**

**zero y-intercept.**

[Figure]

Figure R2. Scatterplots of the determined mileage-based emission factors of (a)

benzene versus toluene, (b) $C_{14}$ aromatics versus toluene, (c) formaldehyde versus toluene, and (d) acetaldehyde versus toluene for gasoline and diesel vehicles. Each data point represents each test vehicle in this study. The green and orange lines are the fitted results for gasoline and diesel vehicle. The black line represents 1:1 ratio, and the shaded areas represent ratio of a factor of 10.

*2.Please review again thoroughly for grammar. A few corrections are:*

*Line 224 "species emitted by vehicles"*

*Line 377 "Comparing gasoline and diesel vehicles,"*

*Line 445. "can be determined"*

*Line 486. "Substantially larger"*

Reply: We thank the reviewer for the comment. We corrected all these comments and checked the grammar throughout the manuscript.

**Reference:**

Alarcon, P., Bohn, B., and Zetzsch, C.: Kinetic and mechanistic study of the reaction of OH radicals with methylated benzenes: 1,4-dimethyl-, 1,3,5-trimethyl-, 1,2,4,5-, 1,2,3,5- and 1,2,3,4-tetramethyl-, pentamethyl-, and hexamethylbenzene, Phys Chem Chem Phys, 17, 13053-13065, 10.1039/c5cp00253b, 2015.

Atkinson, R., and Arey, J.: Atmospheric Degradation of Volatile Organic Compounds, Chemical Reviews, 103, 4605-4638, 10.1021/cr0206420, 2003.

Berndt, T., and Böge, O.: Rate constants for the gas-phase reaction of hexamethylbenzene with OH radicals and H atoms and of 1, 3, 5-trimethylbenzene with H atoms, International Journal of Chemical Kinetics, 33, 124-129, 2001.

de Gouw, J., Middlebrook, A., warneke, C., Goldan, P., Kuster, W., Roberts, J., Fehsenfeld, F., Worsnop, D., Pszenny, A., Keene, W., Marchewka, M., Bertman, S., and Bates, T.: Budget of organic carbon in a polluted atmosphere: Results from the New England Air Quality Study in 2002, Journal of Geophysical Research-Atmospheres, 110, D16305, 10.1029/2004JD005623, 2005.

Gentner, D. R., Worton, D. R., Isaacman, G., Davis, L. C., Dallmann, T. R., Wood, E. C., Herndon, S. C., Goldstein, A. H., and Harley, R. A.: Chemical Composition of Gas-Phase Organic Carbon Emissions from Motor Vehicles and Implications for Ozone Production, Environmental Science & Technology, 47, 11837-11848, 10.1021/es401470e, 2013.

Gentner, D. R., Jathar, S. H., Gordon, T. D., Bahreini, R., Day, D. A., El Haddad, I., Hayes, P. L., Pieber, S. M., Platt, S. M., de Gouw, J., Goldstein, A. H., Harley, R. A., Jimenez, J. L., Prevot, A. S., and Robinson, A. L.: Review of Urban Secondary Organic Aerosol Formation from Gasoline and Diesel Motor Vehicle Emissions, Environ Sci Technol, 51, 1074-1093, 10.1021/acs.est.6b04509, 2017.

Hou, S., and Jiang, X.: Determination of Hydrocarbon Composition in Diesel Oil by Gas Chromatography Mass Spectrometry (in Chinese), Technology & Development of Chemical Industry, 47, 57-58+69, 2018.

Huang, J., Yuan, Z., Duan, Y., Liu, D., Fu, Q., Liang, G., Li, F., and Huang, X.: Quantification of temperature dependence of vehicle evaporative volatile organic compound emissions from different fuel types in China, Science of The Total Environment, 813, 152661, https://doi.org/10.1016/j.scitotenv.2021.152661, 2022.

Liu, W., and Zhang, X.: Determination of Polycyclic Aromatic Hydrocarbons in Diesel with Gas Chromatography - Mass Spectrometry (in Chinene), Guangzhou Chemical Industry, 43, 139-141, 2015.

Qi, L., Zhao, J., Li, Q., Su, S., Lai, Y., Deng, F., Man, H., Wang, X., Shen, X. e., Lin, Y., Ding, Y., and Liu, H.: Primary organic gas emissions from gasoline vehicles in China: Factors, composition and trends, Environmental Pollution, 290, 117984, https://doi.org/10.1016/j.envpol.2021.117984, 2021.

Sha, Q., Zhu, M., Huang, H., Wang, Y., Huang, Z., Zhang, X., Tang, M., Lu, M., Chen, C., Shi, B., Chen, Z., Wu, L., Zhong, Z., Li, C., Xu, Y., Yu, F., Jia, G., Liao, S., Cui, X., Liu, J., and Zheng, J.: A newly integrated dataset of volatile organic compounds (VOCs) source profiles and implications for the future development of VOCs profiles in China, Sci Total Environ, 793, 148348, 10.1016/j.scitotenv.2021.148348, 2021.

Sun, L., Zhong, C., Peng, J., Wang, T., Wu, L., Liu, Y., Sun, S., Li, Y., Chen, Q., Song, P., and Mao, H.: Refueling emission of volatile organic compounds from China 6 gasoline vehicles, Science of The Total Environment, 789, 147883, https://doi.org/10.1016/j.scitotenv.2021.147883, 2021.

Tan, Z., Lu, K., Hofzumahaus, A., Fuchs, H., Bohn, B., Holland, F., Liu, Y., Rohrer, F., Shao, M., Sun, K., Wu, Y., Zeng, L., Zhang, Y., Zou, Q., Kiendler-Scharr, A., Wahner, A., and Zhang, Y.: Experimental budgets of OH, HO$_2$, and RO$_2$ radicals and implications for ozone formation in the Pearl River Delta in China 2014, Atmospheric Chemistry and Physics, 19, 7129-7150, 10.5194/acp-19-7129-2019, 2019.

Tang, G., Sun, J., Wu, F., Sun, Y., Zhu, X., Geng, Y., and Wang, Y.: Organic composition of gasoline and its potential effects on air pollution in North China, Science China Chemistry, 58, 1416-1425, 10.1007/s11426-015-5464-0, 2015.

Wang, H., L. , Jing, S., A. , Lou, S., R. , Hu, Q., Y. , Li, L., Tao, S., K. , Huang, C., Qiao, L., P. , and Chen, C., H.: Volatile organic compounds (VOCs) source profiles of on-road vehicle emissions in China, Sci Total Environ, 607-608, 253-261, 10.1016/j.scitotenv.2017.07.001, 2017.

Wang, X., Tian, Z., and Zhang, Y.: Influence of Fuel Quality on Vehicle Emission and Economic Analysis of Upgrading Fuel Quality in China (in chinese), Bulletin of Chinese Academy of Sciences, 30, 535-541, 10.16418/j.issn.1000-3045.2015.04.013, 2015.

Wang, Z., Yuan, B., Ye, C., Roberts, J., Wisthaler, A., Lin, Y., Li, T., Wu, C., Peng, Y., Wang, C., Wang, S., Yang, S., Wang, B., Qi, J., Wang, C., Song, W., Hu, W., Wang, X., Xu, W., Ma, N., Kuang, Y., Tao, J., Zhang, Z., Su, H., Cheng, Y., Wang, X., and Shao, M.: High Concentrations of Atmospheric Isocyanic Acid (HNCO) Produced from Secondary Sources in China, Environ Sci Technol, 54, 11818-11826, 10.1021/acs.est.0c02843, 2020.

Yue, X., Wu, Y., Hao, J., Pang, Y., Ma, Y., Li, Y., Li, B., and Bao, X.: Fuel quality management versus vehicle emission control in China, status quo and future perspectives, Energy Policy, 79, 87-98, https://doi.org/10.1016/j.enpol.2015.01.009, 2015.

Zhao, Y., Nguyen, N. T., Presto, A. A., Hennigan, C. J., May, A. A., and Robinson, A. L.: Intermediate Volatility Organic Compound Emissions from On-Road Gasoline Vehicles and Small Off-Road Gasoline Engines, Environmental Science & Technology, 50, 4554-4563, 10.1021/acs.est.5b06247, 2016.

---

## Author Comment (AC2)

**Response to Reviewers**

**Reviewer #**2

Wang et al. present an analysis of VOC emissions measured from vehicle dynamometer 3 testing for vehicles designed under different emission standards (China I - IV). The 4 authors evaluate total and speciated VOC emissions from both gasoline, diesel, and 5 LPG under a variety of conditions (cold start, warm start, speed, etc). The authors 6 7 detail the different emission factors between each vehicle, and observe a distinct 8 difference between the OVOCs emitted by gasoline and diesel engines. The latter produces significantly higher fraction of OVOCs than by gasoline, which appears to be 9 10 at least partly associated with the pollution control technology.

*I found the paper to be very well-written, well-reasoned, and full of good information. I appreciated the study as a nice piece of work describing fossil fuel emissions from motor vehicles in China.*

My only substantive comment is that I don't have a sense of the fuel composition and how this might contribute to the high OVOCs observed in diesel exhaust. And how do the OVOC emissions compare against diesel exhaust studies reported elsewhere? Gentner et al. (2013) also see elevated OVOC emissions in diesel compared to gasoline. Are these differences comparable to what is observed here, or is there something different between the aftertreatment or fuels that could contribute to any differences?

Reply: We would like to thank the reviewer for the insightful comments, which helped us tremendously in improving the quality of our work. Fuel composition is one of determining factor for VOCs emissions from vehicles (Gentner et al., 2017). We conducted some literature review and added a discussion in Section 1 in the Supplement to provide some information about chemical compositions of gasoline and diesel fuel in China. We also appreciate the reviewer for providing the useful reference. We added some discussions in the Section 3.1 to compare with the result of them.

We also used the fractions of OVOCs in total VOC emissions to compare against
diesel exhaust studies reported elsewhere (Fig.12). If only considering carbonyls among
various types of OVOCs measured by PTR-ToF-MS, the OVOC fractions determined in this study are more comparable with previous studies. We also discussed higher
OVOC emissions in diesel vehicles and impact on after-treatment devices, please find
the response to individual comments below.

The sentence in the Section 2.1 (line 126-127) is modified to:

The detailed information for test vehicles is summarized in Sect. 1 in the

Supplement, Table S2 and Table S3.

The Section 1 in the Supplement is modified to:

Fuel composition is one of determining factor for VOCs emissions from 37 vehicles (Gentner et al., 2017). The gasoline fuel used in China is mainly comprised 38 of C4-C7 hydrocarbons. The chemical compositions of gasoline fuel are alkanes 39 (55%-62%), alkenes (12%-17%), aromatics (27%-32%), and methyl tert-butyl 40 ether (MTBE, 1%-4%) (Tang et al., 2015;Sun et al., 2021;Qi et al., 2021;Huang et 41 al., 2022). Heavy hydrocarbons, namely C8-C10 alkanes and aromatics, 42 contributed most in diesel fuel. The chemical compositions of diesel are alkanes 43 (70%-79%), alkenes (1%-7%), and aromatics (21%-25%) (Wang et al., 2015; Yue 44 et al., 2015; Hou and Jiang, 2018; Liu and Zhang, 2015). Gasoline and diesel fuel 45 are summer blends, and the gasoline fuel does not content ethanol in this study. 46

The sentences in the Section 3.1 (line 284-289) are modified to:

As the largest OVOCs emitted from gasoline vehicles  $(4.6 \pm 5.1 \text{ mg} \cdot \text{km}^{-1})$ , 49 methanol is found to be the only common OVOC species, with lower emission 50 factors from diesel vehicles than gasoline vehicles. The emission factor of other 51 OVOCs (e.g., formaldehyde, acetone) from diesel vehicles are higher than gasoline 52 vehicles, which is consistent with previous results (Gentner et al., 2013).

*Comments*:

Line 70-71 Based on the reference, I presume that the authors are specifically noting
the decline of VOCs in urban regions in China? For clarity, I would suggest re-writing
this sentence to say "Furthermore, VOC emission significantly decreased in China due
to stricter emission standards."

Reply: We thank the reviewer for the comment. We corrected this sentence as **"Furthermore, VOC emissions from vehicles significantly decreased in China due**to stricter emission standards (Liu et al., 2017;Sha et al., 2021)".

2. Line 76: Could the authors provide some context on the China VI emission standard?
I recognize that the standard is dependent on power ranges, but a few sentences on
VOC emissions at max power output would be useful. This would also be useful in the
methods (lines 112 - 122) to give readers context as to what the China I - IV standards
represent in terms of VOC emissions.

Reply: We thank the reviewer for the comment. China VI emission standard is the newest emission standard for vehicles. Limits on exhaust emissions of gasoline vehicles are tightened by 30% to 50% from China V to China VI for different pollutants (Lyu et al., 2020). China VI emission standard is mainly reflected in the requirements for the emission limits of pollutants (e.g. CO, NOX, THC, etc.). In response to the reviewer's question, power ranges and max power outputs are not directly reflected in China VI emission standard, and rarely reported in previous studies.

We added a discussion in Section 1 in the Supplement to provide some 75 information about the China VI emission standard of vehicles. We added Table S7 with 76 the different emission standards (China I - China VI) for light-duty vehicles (LDV) in 77 gasoline and diesel as fuel, and Table S8 for heavy-duty diesel engines (HDDE) in 78 different emission standards (China I - China V). We also added some discussion in the 79 Section 2.1 to provide the averaged fractions of gasoline and diesel vehicles with 80 different emission standards for the vehicle fleet in China, which is shown in the Table 81 82 S1 in the revised manuscript (Table S6 in the original manuscript).

The sentences in the Section 1(line 76-78) are modified to:

The emission limits for various air pollutants emitted by vehicles are significantly lower under the China VI emission standard (see details in the Supplement) (Wu et al., 2017).

The Section 1 in the Supplement is modified to:

The limits and measurement methods for emissions of light-duty vehicles 88 (GB18352.6-2016; known as the China VI standard) are introduced in the recent 89 90 years in China, which applies to light-duty vehicles by gasoline or diesel as the fuel. The China VI emission standard continued the EU standard system as the 91 reference with various regulation details integrated from US emissions standards 92 93 (Lyu et al., 2020). Vehicle emission limits are significantly lower for the China VI standard (Table S7, Table S8). For example, limits on gasoline vehicle exhaust 94 emissions were tightened by 30 to 50% from China V to China VI, and a new 95 particulate number (PN) limit was added in gasoline vehicles (Lyu et al., 2020). 96 The sentence in the Section 2.1 (line 123-124) is modified to: 97 The averaged fractions of gasoline and diesel vehicles with different emission 98 standards for the vehicle fleet in China are shown in Table S1 (MEEPRC, 2019;Li 99 100 et al., 2021). 101 3. Line 81: Would suggest modifying "group" to say "class of compounds" 102 103 Reply: We replaced "group" with "class of compounds". 104 4. Line 80-83: Are the authors primarily discussing VOC measurements from 105 dynamometer studies, or tunnel studies, or ambient studies? I think the distinction 106 matters given that results from laboratory, tunnel, or ambient measurements can be 107 interpreted differently given differences in co-emitted sources that can convolute the 108 109 measured signal from tailpipe emissions Reply: We thank the reviewer for the comment. We modified this sentence in the 110 111 Section 1 to make it clearly in different vehicle measurement methods. The sentence in the Section 1(line 82-86) is modified to: 112 Oxygenated volatile organic compounds (OVOCs) were found to be an 113 important class of compounds in vehicle exhausts, accounting for more than 50% 114 115 of the total VOC emissions for diesel vehicles from both chassis dynamometer tests (Schauer et al., 1999; Mo et al., 2016) and on-road mobile measurements (Yao et al., 2015).

Lines 268-271: Are there also differences in the aftertreatment that might lead to
higher OVOC emissions? The authors note the temperature of the device at line 240,
and I'm curious if previous work has looked at VOC speciation under different
aftertreatment conditions.

Reply: We thank the reviewer for the comment. After-treatment devices in 123 vehicles have been improved associated with the upgrading of emission standards. 124 According to the #8 comment of the reviewer, our results (Fig. 7c-d in the revised 125 manuscript) actually can answer the question of the reviewer in this comment. The two 126 graphs show that the chemical compositions of VOC emissions are comparable between 127 different emission standards for both gasoline and diesel vehicles (R=0.98 and 0.89), 128 129 indicating after-treatment devices may not affect the relative fraction of VOC components. We added some description and reference in the section 3.2, and added a 130 discussion in Section 1 in the Supplement to provide some information about the after-131 132 treatment devices in gasoline and diesel vehicles.

The sentence in the Section 3.2 (line 363-366) is modified to:

Fig. 7c-d show that the chemical compositions of VOC emissions are comparable between different emission standards for both gasoline and diesel vehicles (R=0.98 and 0.89), indicating after-treatment devices may not affect the relative fractions of VOC components.

The sentence in the Section 3.2 (line 379-383) is modified to:

These results indicate the after-treatment device for diesel vehicles (see Sect.
in the Supplement for details.) may effectively reduce emissions of some heavier
VOC species, though the after-treatment devices do not aim for VOCs control
(Gentner et al., 2017).

The Section 1 in the Supplement is modified to:

144After-treatment devices commonly used in light-duty gasoline vehicles are145three-way catalyst (TWC) and gasoline particulate filter (GPF). They have been improved with the upgrading of emission standard. For diesel vehicles, typical 146 after-treatment devices include diesel oxidation catalyst (DOC), diesel particulate 147 filter (DPF), and selective catalyst reduction (SCR) (Zhou et al., 2019;Lyu et al., 148 2020; Shen et al., 2021). The diesel vehicles for China III or prior do not have any 149 after-treatment devices. Light-duty-diesel-truck (LDDT) used DOC and 150 DOC+DPF as after-treatment devices in China IV and V diesel vehicles, 151 respectively. SCR devices are mainly used for heavy-duty-diesel-truck (HDDT) 152 with China IV and V as after-treatment devices. 153

6. Figure 1: It would be useful to see the acronyms (LDDT, MDDT, HDDT, and BUS)
defined in the caption as a reminder to the reader.

Reply: We thank the reviewer for the comment. We added some description in 158 the caption of Fig. 1about the acronyms (LDDT, MDDT, HDDT, and BUS). We also 159 checked throughout the manuscript, and corrected the caption of Fig.2.

The caption of Fig.1 is modified to:

Figure 1. Real-time concentrations of acetaldehyde, acetone, benzene, toluene, and CO2 for (a) a gasoline vehicle with emission standard of China I and (b) a light-duty diesel vehicle (LDDV) with emission standard of China IV. The two vehicles were both cold started. The gray shadows represent the speed of the vehicles on the chassis dynamometer.

The caption of Fig.2 is modified to:

Figure 2. The determined average mileage-based emission factors (mg·km-1) 167 for (a) benzene, (b) toluene, (c) acetaldehyde, and (d) acetone for vehicles with 168 169 different emission standards. The numbers above the top axis represent the number of all experiments (including multiple measurements for individual test 170 vehicle) for each emission standard. LDDT, MDDT, HDDT, and BUS represent 171 light-duty-diesel-truck, middle-duty-diesel-truck, heavy-duty-diesel-truck, and 172 bus, respectively. Error bars represent standard deviations of emission factors for 173 the specific emission standard. 174

- 176 7. Title of Section 3.2: The title doesn't quite reflect the discussion that follows. Might I
- 177 suggest "Analysis of PTR-ToF-MS mass spectra to evaluate VOC speciation"?
- 178 Reply: We corrected this title in "Analysis of PTR-ToF-MS mass spectra to179 evaluate VOCs speciation".
- 180

8. Lines 320-323: This is a nice result, and partially addresses my question at lines 268182 271. Could the authors point to this figure and discussion to demonstrate that the
changes to the VOC distribution isn't significantly different between cold start and
normal operation?

Reply: We thank the reviewer for the comment. After-treatment devices have 186 been improved with the upgrading of emission standard. Our results (Fig. 7c-d in the 187 revised manuscript) show that the chemical compositions of VOC emissions are 188 comparable between different emission standards in gasoline and diesel vehicles 189 (R=0.98 and 0.89), indicating after-treatment devices may not affect the relative 190 fraction of VOC components.

Cold start is a major emission source of gasoline vehicles, which occurs after 191 several hours of non-operation of vehicles (Gentner et al., 2017;George et al., 2015). 192 Our results (Fig. 7a-b in the revised manuscript) demonstrate that variation behaviors 193 are similar for different species and thus chemical compositions of VOC emissions are 194 comparable between different start conditions. As cold start emissions are richer in 195 unburned fuel than other hot-running conditions, the observation in Fig. 7a-b also infer 196 that unburned fuel are the major contributor for vehicle exhaust emissions, which has 197 been previously shown in California, US (Gentner et al., 2013). We added some 198 discussions in 3.2 and Section 1 in the Supplement to provide some information about 199 cold start in gasoline and diesel vehicles. 200

The sentence in the Section 3.2 (line 247-249) is modified to:

It might be a combined effect of cold engine and operation temperature of
the after-treatment device (Gentner et al., 2017;George et al., 2015).

**The sentences in the Section 3.2 (line 345-353) are modified to:**

We observe strong correlation between emission factors from cold start and 205 206 hot start tests (R=0.99 and 0.92) and generally consistent ratios between cold start and hot start for different types of VOC species for both gasoline and diesel 207 vehicles, indicating that variation behaviors are similar for different species and 208 209 thus chemical compositions of VOC emissions are comparable between different start conditions. As cold start emissions are richer in unburned fuel than other 210 hot-running conditions, the observation in Fig. 7a-b also infer that unburned fuel 211 are the major contributor for vehicle exhaust emissions, which has been previously 212 213 shown in California, US (Gentner et al., 2013).

The Section 1 in the Supplement is modified to:

Cold start, which occurs after several hours of nonoperation for vehicles 215 216 (Drozd et al., 2016), is a major source of emissions for gasoline vehicles and have greater emissions due to two issues: (1) low engine temperatures lead to incomplete 217 combustion that allow non/partially combusted fuel compounds to exit engine 218 219 cylinders. (2) Effective operation of the catalytic converter requires a warm-up period to reach sufficient catalyst operating temperatures (Gentner et al., 220 2017;George et al., 2015). Due to diesel emissions have emphasized control of 221 222 primary PM2.5 and NOX emissions, the after-treatment devices of diesel vehicles (e.g. DOC, DPF, SCR etc.) do not aim for VOCs control. 223

9. Lines 424-425: I like the discussion in this section on using the aromatics to delineate
between diesel and gasoline. I agree with the authors that these ratios might be difficult
to assess in the ambient owing to additional sources of aromatics (e.g. solvent emissions)
and secondary production of formaldehyde and acetaldehyde. Are there any unique
masses, with high enough signal in ambient air, that could be used to more definitively
separate gasoline vs diesel emissions? I also wonder if ratios to CO or other
combustion markers might be insightful.

Reply: We thank the reviewer for the comment. Per the reviewer's comment, we have not found any other unique masses with high enough signals in gasoline or dieselvehicles that can be used for distinguish the two types of vehicles.

Furthermore, we added a Figure (Fig. S11b) with the emission ratios to CO (ppb·ppm-1) between gasoline and diesel vehicles. The result (slope=0.16) is similar to the plot of emission factors between gasoline and diesel vehicles. A limited number of VOC species, including C6-C10 aromatics are associated with higher emission ratios from gasoline vehicles, whereas the obtained emission ratios of most VOC species emitted from diesel vehicles are higher, especially most OVOC species.

The sentences in the Section 3.3 (line 405-408) are modified to:

Generally, similar variability is obtained except the determined slope of the data points, with higher slopes determined from the scatterplot based on fuelbased emission factor (0.19 versus 0.15). The emission ratios to CO between gasoline and diesel vehicles (Fig. S11b) show similar results.

Figure S11. Scatterplot of (a) the determined average fuel-based emission factors (mg·kgfuel-1) and (b) the emission ratios to CO (ppb·ppm-1) of VOCs between gasoline and diesel vehicles. Each data point indicates a VOC species measured by PTR-ToF-MS. The blue line is the fitted result for all data points. The black line represents 1:1 ratio, and the shaded areas represent ratios of a factor of 10 and 100.

10. Figure S6. The intercomparisons are nice for the fast time-resolution systems, but there are significant differences between the GC and PTR for toluene - is this due to differences in sampling techniques (e.g., grab sampling artifacts vs real-time sampling), or something due to fragmentation in the PTR to produce a signal at m/z 93? I believe the other reviewer also commented on this, and I agree that some explanation is warranted here.

Reply: We thank the reviewer for the comment. In replying to the comments from 260 the two reviewers, we found an issue in data analysis for preparing the original 261 manuscript. The alignment of data points between offline canister-GC-MS/FID and 262 PTR-ToF-MS for several gasoline vehicles and diesel vehicles was not correct. We have 263 modified the corresponding data points, and added comparison of C8 aromatics between 264 two measurements (Fig. S6), obtaining generally consistent results, considering large 265 variations of VOC emissions for driving conditions and the difficulty to control the fill 266 time for canisters. We also revised related figures (Fig. 12 and Fig. S12) and description 267 in the manuscript on the fractions of OVOCs in total VOC emissions in various types 268 269 of vehicles. These modifications do not change any conclusion in the manuscript.

The sentence in the Section 2.3 (line 194-197) is modified to:

We compared emission factors from PTR-ToF-MS and the offline canister-GC-MS/FID (Fig. S6c-d), obtaining generally consistent results, considering the large variation of VOC emissions for driving conditions and the difficulty to control the fill time for canisters.

---

## Referee Report (RR1)

Wang et al. present a revised manuscript that addresses many of the comments in my previous review. I appreciate the authors works, and I am satisfied by the responses. Overall, I support publication. There is one remaining comment that I would appreciate if the authors could address, as I believe it will help clarify a question that I posed in my initial review. I've also made a number of comments on the new content in the SI that I think will help to clarify the material.

**Main Comment**

Lines 359 - 362 : This new text is confusing, and I believe this was added to address comments 5 and 8 of my previous review. Admittedly, my initial questions may have not been clear. In my previous review, I asked whether Figures 7a-b could provide information about the effects of the after treatment process on VOC profiles. I presumed that the comparison between cold-start emissions and hot-start emissions were sufficient to address this question. Really, my aim was to hear more from the authors about the source of VOC emissions, and I think the authors now effectively address this at lines 346-349 with the discussion of unburnt fuel.

In the new text, the authors point to Figures 7c-d to argue that the after treatment process has little effect on VOC profiles. I do not agree that these panels provide strong evidence for this conclusion. Figure 7d shows significant scatter, and the correlation coefficient derived from these data seem to be driven by a select number of high emission VOCs. Furthermore, after re-reading this section, this new text conflicts with the statement at lines 375-377, which suggest that the "after-treatment device for diesel vehicles may effectively reduce emissions of some heavier VOC species."

I think this can be resolved by simply removing the text at lines 359 - 362. Ultimately, I don't think this text adds much to the discussion. I appreciate the efforts by the authors to address my comments.

**Comments on Supplement:**

Lines 46 - 55 in the Supplement: This information is really useful to the reader in order to understand how the emission control technologies have changed under different standards. I think this section should be elevated to the main text. A good place for this could be at line 121 after the description of the LPG vehicles.

Line 26: Please add "the" between "of" and "determining"

Line 35: "Content" should be "contain"

Line 37 - 39: Wording is a little awkward, would suggest rephrasing as "… have been recently introduced in China, which applies to light-duty vehicles using gasoline and diesel fuel"

Line 49: Would suggest re-wording "upgrading of emission standard" to say "stricter emission standards"

Line 74: I believe "cycle" should be plural

Line 96 - 97: This reads awkwardly. I suggest revising to read " Here, the limit of detection for VOC mixing ratios were calculated and applied to estimate the limit of detection for emission factors"

Line 98: Would suggest removing "kind of"

Line 99-102: I don't follow what is written here - are the authors saying that the mass spectra is below the limit of detection for most measurements?  I don't fully understand why one vehicle is used here to infer the LOD/Signal ratio here.

Line 106 - 112: I'm not sure why the discussion of $C_{16}H_{22}O_4H$ is included here. If the authors do not believe this compound is a part of the tailpipe emissions, then I would remove this from the discussion. If this compound is of interest for other reasons (i.e., some sort of plasticizer?) then I believe the authors should provide some discussion. But to my eye, this seems to be a part of the dynamometer system and can be reasonably discarded.

Line 144-146: This reads a bit awkwardly - I would suggest saying "The average rate constant for C14 aromatics has not been reported, so we assume a rate constant similar to representative C12 aromatics"

---

## Author Response (AR2)

**Response to reviewers' comments**

**Reviewer #1**

*Overview*

*The authors have made many significant improvements to the manuscript. Both the text and the figures have improved significantly. However, some important edits largely related to the presentation and interpretation of results should be still addressed.*

Reply: We would like to thank the reviewer for the insightful comments, which helped us tremendously in improving the quality of our work. Please find the response to individual comments below.

*1. The discussion of the consistency in composition across emissions standards is problematic. For gasoline vehicles the authors' results are consistent with previous findings1–3, which should be referenced on line 358. For diesel vehicles, the author's interpretation of the data is either misleading or in error, based on Figure 7d. The blue line showing the fit to the data passes through very few of the points. (The opposite is true for gasoline, in which the line seems to pass through the spread of the data points). For diesel, it thus appears that either a small number of compounds heavily impact the fit, or the fit is somehow in error. Thus the authors should not claim that diesel emissions are not changing with emissions standards, because it would appear only a few major compounds are not varying, but the others could vary significantly. The argument based on the questionable R-squared value is not acceptable in this case, because the distribution of the data about the fit is very far from normal.*

Reply: We thank the reviewer for the comment. We agree with the reviewer that the discussions of the consistency in composition across emissions standards for diesel vehicles are not appropriate. We have revised the text added at lines 359-362 in the latest version manuscript. We appreciate the reviewer for providing these useful references.

The sentences in the Section 3.2 (line 360-366) are modified to:

**Fig. 7c show that the chemical compositions of VOC emissions are**

**comparable between different emission standards for abundant VOC species from gasoline vehicles, indicating after-treatment devices may not affect the relative fractions of VOC components for gasoline vehicles (Drozd et al., 2019;Lu et al., 2018;Zhao et al., 2017). In comparison, the results between different emission standards for diesel vehicles (Fig. 7d) are somewhat larger than in gasoline vehicles.**

*2. A similar problem exists for the cold-start vs. hot-start emissions. In general the argument that unburned fuel dominates the emissions for gasoline vehicles seems consistent with previous literature, although the BTEX compounds are a notable exception with different ratios in emissions vs. fuel4, which should be noted. How can the authors make the same claim for diesel, when such a large fraction of the diesel emissions is reported to be OVOCs? Diesel fuel is not more than 50% OVOC, so despite any data analysis here, the major claim of the manuscript concerning OVOC fractions in emissions, does not allow for the emissions to be comprised of unburned fuel. This statement must be removed, and some alternative explanation for the correlation of cold-start and hot-start emissions must be suggested. Perhaps the OVOC may be derived from particular fuel components, yet still the authors have no information on fuel composition, the largest fault in this study.*

Reply: We thank the reviewer for the insightful comment. We have revised description in the manuscript on the unburned fuel dominates the emissions for gasoline vehicles. For diesel vehicles, we have removed related description. Explanation for the correlation of cold-start and hot-start emissions have described in Lines 339-346 of the revised manuscript. The information on fuel composition had been added in the Sect. 1 in the supplement in the last version manuscript, and the high emissions of OVOCs from diesel vehicles may be related to combustion processes in diesel vehicles, with more excess air (i.e., under overall fuel-lean conditions) into combustion cylinder (Gentner et al., 2017), we had claim about it in the lines 293-297 in the revised manuscript.

The sentences in the Section 3.2 (line 346-350) are modified to:

**As cold start emissions are richer in unburned fuel than other hot-running conditions (Gentner et al., 2017) and the after-treatment devices aim for VOCs control for gasoline vehicles, the strong correlation and significantly lower slope than unity in Fig. 7a infer that unburned fuel are the major contributor for exhaust emissions of gasoline vehicles, which has been previously shown in California, U.S. (Gentner et al., 2013).**

*3. Finally, the manuscript needs to be carefully reviewed for grammar and syntax again. A couple egregious issues are noted below.*

*Line 263 "Intestinally, the emission factors of the representative VOC species are highest for China II gasoline vehicles rather than China I vehicles, coincidence with largest mileage of the test vehicles. This sentence should be changed to: "The emission factors of the representative VOC species are highest for China II gasoline vehicles rather than China I vehicles, which can be explained by the China II vehicles having the highest mileage of the test vehicles."*

*Line 445 "The remarkable larger emission factors of C14 aromatics from diesel vehicles suggest that diesel vehicles can be a significant or even predominated source for higher molecular aromatics" This sentence should be changed to: "The significantly higher emission factors of C14 aromatics from diesel vehicles suggest that diesel vehicles can be a significant or even dominant source for higher molecular-weight aromatics"*

We thank the reviewer for the comment. We corrected all these comments and checked the grammar and syntax throughout the manuscript.

The sentence in line 272-274 in the revised manuscript is modified to:

**The emission factors of the representative VOC species are highest for China II gasoline vehicles rather than China I vehicles, which can be explained by the China II vehicles having the highest mileage of the test vehicles.**

The sentence in line 453-455 in the revised manuscript is modified to:

**The significantly higher emission factors of $C_{14}$ aromatics from diesel vehicles suggest that diesel vehicles can be a significant or even dominant source for higher molecular-weight aromatics.**

*References*

*(1) Lu, Q.; Zhao, Y.; Robinson, A. L. Comprehensive Organic Emission Profiles for Gasoline, Diesel, and Gas-Turbine Engines Including Intermediate and Semi-Volatile Organic Compound Emissions. Atmos. Chem. Phys. Discuss. 2018, 18, 1–28.*

*(2) Drozd, G. T. G. T.; Zhao, Y.; Saliba, G.; Frodin, B.; Maddox, C.; Chang, M.-C. O. O.; Maldonado, H.; Sarder, S.; Weber, R. J. R. J.; Robinson, A. L.; et al. Detailed Speciation of Intermediate Volatility and Semivolatile Organic Compound Emissions from Gasoline Vehicles: Effects of Cold-Starts and Implications for Secondary Organic Aerosol Formation. Environ. Sci. Technol. 2019, 53 (3), 1706–1714.*

*(3) Zhao, Y.; Saleh, R.; Saliba, G.; Presto, A. A.; Gordon, T. D.; Drozd, G. T.; Goldstein, A. H.; Donahue, N. M.; Robinson, A. L. Reducing Secondary Organic Aerosol Formation from Gasoline Vehicle Exhaust: Precursors and NOx Effects. Proc. Natl. Acad. Sci. 2017, 114 (27), 6984–6989.*

*(4) Drozd, G. T.; Zhao, Y.; Saliba, G.; Frodin, B.; Maddox, C.; Weber, R. J.; Chang, M.-C. O. C. O.; Maldonado, H.; Sardar, S.; Robinson, A. L.; et al. Time Resolved Measurements of Speciated Tailpipe Emissions from Motor Vehicles: Trends with Emission Control Technology, Cold Start Effects, and Speciation. Environ. Sci. Technol. 2016, 50 (24), 13592–13599.*

**Reviewer #2**

*Wang et al. present a revised manuscript that addresses many of the comments in my previous review. I appreciate the authors works, and I am satisfied by the responses. Overall, I support publication. There is one remaining comment that I would appreciate if the authors could address, as I believe it will help clarify a question that I posed in my initial review. I've also made a number of comments on the new content in the SI that I think will help to clarify the material.*

Reply: We would like to thank the reviewer for the insightful comments, which helped us tremendously in improving the quality of our work. Please find the response to individual comments below.

*Main Comment*

*Lines 359 - 362: This new text is confusing, and I believe this was added to address comments 5 and 8 of my previous review. Admittedly, my initial questions may have not been clear. In my previous review, I asked whether Figures 7a-b could provide information about the effects of the after treatment process on VOC profiles. I presumed that the comparison between coldstart emissions and hot-start emissions were sufficient to address this question. Really, my aim was to hear more from the authors about the source of VOC emissions, and I think the authors now effectively address this at lines 346-349 with the discussion of unburnt fuel.*

*In the new text, the authors point to Figures 7c-d to argue that the after treatment process has little effect on VOC profiles. I do not agree that these panels provide strong evidence for this conclusion. Figure 7d shows significant scatter, and the correlation coefficient derived from these data seem to be driven by a select number of high emission VOCs. Furthermore, after rereading this section, this new text conflicts with the statement at lines 375-377, which suggest that the "after-treatment device for diesel vehicles may effectively reduce emissions of some heavier VOC species."*

*I think this can be resolved by simply removing the text at lines 359 - 362. Ultimately, I don't think this text adds much to the discussion. I appreciate the efforts by the authors*

*to address my comments.*

Reply: We thank the reviewer for the comment. We agree with the reviewer that the discussions of the consistency in composition across emissions standards for diesel vehicles are not appropriate. We have revised the text added at lines 359-362 in the latest version manuscript.

The sentences in the Section 3.2 (line 360-366) are modified to:

**Fig. 7c show that the chemical compositions of VOC emissions are comparable between different emission standards for abundant VOC species from gasoline vehicles, indicating after-treatment devices may not affect the relative fractions of VOC components for gasoline vehicles (Drozd et al., 2019;Lu et al., 2018;Zhao et al., 2017). In comparison, the results between different emission standards for diesel vehicles (Fig. 7d) are somewhat larger than in gasoline vehicles.**

*Comments on Supplement:*

*1. Lines 46 - 55 in the Supplement: This information is really useful to the reader in order to understand how the emission control technologies have changed under different standards. I think this section should be elevated to the main text. A good place for this could be at line 121 after the description of the LPG vehicles.*

Reply: We thank the reviewer for the comment. We have removed this section in the Supplement, and added them in the Section 2.1.

The sentences in the Section 2.1 (line 120-129) in the revised manuscript are modified to:

**After-treatment devices commonly used in light-duty gasoline vehicles are three-way catalyst (TWC) and gasoline particulate filter (GPF) (Lyu et al., 2020). They have been improved with the stricter emission standards. For diesel vehicles, typical after-treatment devices include diesel oxidation catalyst (DOC), diesel particulate filter (DPF), and selective catalyst reduction (SCR) (Zhou et al., 2019;Lyu et al., 2020;Shen et al., 2021). The diesel vehicles for China III or prior**

**do not have any after-treatment devices. Light-duty-diesel-truck (LDDT) used DOC and DOC+DPF as after-treatment devices in China IV and V diesel vehicles, respectively. SCR devices are mainly used for heavy-duty-diesel-truck (HDDT) with China IV and V as after-treatment devices.**

*2. Line 26: Please add "the" between "of" and "determining"*

Reply: We add "the" between "of" and "determining".

*3. Line 35: "Content" should be "contain"*

Reply: We replaced "content" with "contain".

*4. Line 37 - 39: Wording is a little awkward, would suggest rephrasing as "… have been recently introduced in China, which applies to light-duty vehicles using gasoline and diesel fuel"*

Reply: We thank the reviewer for the comment. The sentence in the 37-39 in the supplement is modified to:

**The limits and measurement methods for emissions of light-duty vehicles (GB18352.6-2016; known as the China VI standard) have been recently introduced in China, which applies to light-duty vehicles using gasoline and diesel fuel.**

*5. Line 49: Would suggest re-wording "upgrading of emission standard" to say "stricter emission standards*

Reply: We replaced "upgrading of emission standard" with "stricter emission standards".

*6. Line 74: I believe "cycle" should be plural*

Reply: We replaced "cycle" with "cycles".

*7. Line 96 - 97: This reads awkwardly. I suggest revising to read " Here, the limit of detection for VOC mixing ratios were calculated and applied to estimate the limit of detection for emission factors"*

Reply: We thank the reviewer for the comment. The sentence in the 86-88 in the Supplement is modified to:

**Here, the limit of detection for VOC mixing ratios were calculated and applied to estimate the limit of detection for emission factors.**

*8. Line 98: Would suggest removing "kind of"*

Reply: We removed "kind of".

*9. Line 99-102: I don't follow what is written here - are the authors saying that the mass spectra is below the limit of detection for most measurements? I don't fully understand why one vehicle is used here to infer the LOD/Signal ratio here.*

Reply: We thank the reviewer for the comment. In this section, due to the large number of ions measured in the mass spectra, we need to consider whether the corresponding emission factors of all ions are effective. Therefore, we take a China V gasoline vehicle (the emission factors may be sufficiently lower) as an example to calculate the ratio of the emission factor to the limit of detection for emission factor.

*10. Line 106 - 112: I'm not sure why the discussion of C16H22O4H is included here. If the authors do not believe this compound is a part of the tailpipe emissions, then I would remove this from the discussion. If this compound is of interest for other reasons (i.e., some sort of plasticizer?) then I believe the authors should provide some discussion. But to my eye, this seems to be a part of the dynamometer system and can be reasonably discarded.*

Reply: We thank the reviewer for the comment. We have removed this section in the Section 3.2 in the revised manuscript, and revised this section in the Supplement to give an explanation if anyone is interested in this.

The sentences in the 96-100 in the Supplement are modified to:

**It should be noted that the signals of $C_{16}H_{22}O_4H$ (m/z=279) were higher during the tests based on determined emission factors. However, we suspect that it may be emitted artifacts from the sampling or dilution system as it mainly showed higher signals in the latter period of each test when sampling materials absorb more heat from vehicle exhausts (Fig. S12), and thus it is not included in Fig. 5.**

*11. Line 144-146: This reads a bit awkwardly - I would suggest saying "The average rate constant for C14 aromatics has not been reported, so we assume a rate constant similar to representative C12 aromatics"*

Reply: We thank the reviewer for the comment. The sentence in the 132-134 in the Supplement is modified to:

**The average rate constant for $C_{14}$ aromatics has not been reported, so we assume a rate constant similar to representative $C_{12}$ aromatics.**

**Reference:**

Drozd, G. T., Zhao, Y., Saliba, G., Frodin, B., Maddox, C., Oliver Chang, M. C., Maldonado, H., Sardar, S., Weber, R. J., Robinson, A. L., and Goldstein, A. H.: Detailed Speciation of Intermediate Volatility and Semivolatile Organic Compound Emissions from Gasoline Vehicles: Effects of Cold-Starts and Implications for Secondary Organic Aerosol Formation, Environ Sci Technol, 53, 1706-1714, 10.1021/acs.est.8b05600, 2019.

Gentner, D. R., Worton, D. R., Isaacman, G., Davis, L. C., Dallmann, T. R., Wood, E. C., Herndon, S. C., Goldstein, A. H., and Harley, R. A.: Chemical Composition of Gas-Phase Organic Carbon Emissions from Motor Vehicles and Implications for Ozone Production, Environmental Science & Technology, 47, 11837-11848, 10.1021/es401470e, 2013.

Gentner, D. R., Jathar, S. H., Gordon, T. D., Bahreini, R., Day, D. A., El Haddad, I., Hayes, P. L., Pieber, S. M., Platt, S. M., de Gouw, J., Goldstein, A. H., Harley, R. A., Jimenez, J. L., Prevot, A. S., and Robinson, A. L.: Review of Urban Secondary Organic Aerosol Formation from Gasoline and Diesel Motor Vehicle Emissions, Environ Sci Technol, 51, 1074-1093, 10.1021/acs.est.6b04509, 2017.

Lu, Q., Zhao, Y., and Robinson, A. L.: Comprehensive organic emission profiles for gasoline, diesel, and gas-turbine engines including intermediate and semi-volatile organic compound emissions, Atmospheric Chemistry and Physics, 18, 17637-17654, 10.5194/acp-18-17637-2018, 2018.

Lyu, M., Bao, X., Zhu, R., and Matthews, R.: State-of-the-art outlook for light-duty vehicle emission control standards and technologies in China, Clean Technologies and Environmental Policy, 22, 757-771, 10.1007/s10098-020-01834-x, 2020.

Shen, X., Lv, T., Zhang, X., Cao, X., Li, X., Wu, B., Yao, X., Shi, Y., Zhou, Q., Chen, X., and Yao, Z.: Real-world emission characteristics of black carbon emitted by on-road China IV and China V diesel trucks, Science of The Total Environment, 799, 149435, https://doi.org/10.1016/j.scitotenv.2021.149435, 2021.

Zhao, Y., Saleh, R., Saliba, G., Presto, A. A., Gordon, T. D., Drozd, G. T., Goldstein, A. H., Donahue, N. M., and Robinson, A. L.: Reducing secondary organic aerosol formation from gasoline vehicle exhaust, Proc Natl Acad Sci U S A, 114, 6984-6989, 10.1073/pnas.1620911114, 2017.

Zhou, H., Zhao, H., Hu, J., Li, M., Feng, Q., Qi, J., Shi, Z., Mao, H., and Jin, T.: Primary particulate matter emissions and estimates of secondary organic aerosol formation potential from the exhaust of a China V diesel engine, Atmospheric Environment, 218, 116987, https://doi.org/10.1016/j.atmosenv.2019.116987, 2019.